# Skilful seasonal forecasts of streamflow over Europe?

Louise Arnal[1,2], Hannah L. Cloke[1,3,4,5], Elisabeth Stephens[1], Fredrik Wetterhall[2], Christel Prudhomme[2,6,7], Jessica Neumann[1], Blazej Krzeminski[2] and Florian Pappenberger[2]

[1]Department of Geography and Environmental Science, University of Reading, RG6 6AB, United Kingdom
[2]European Centre for Medium-Range Weather Forecasts, Shinfield Park, Reading, RG6 9AX, United Kingdom
[3]Department of Meteorology, University of Reading, RG6 6BB, United Kingdom
[4]Department of Earth Sciences, Uppsala University, Uppsala, SE-752 36, Sweden
[5]Centre of Natural Hazards and Disaster Science, CNDS, Uppsala, SE-752 36, Sweden
[6]Department of Geography, Loughborough University, Loughborough, LE11 3TU, United Kingdom
[7]NERC Centre for Ecology & Hydrology, Wallingford, OX10 8BB, United Kingdom

*Correspondence to*: Louise Arnal (l.l.s.arnal@pgr.reading.ac.uk; louise.arnal@ecmwf.int)

**Abstract.** This paper considers whether there is any added value in using seasonal climate forecasts instead of historical meteorological observations for forecasting streamflow on seasonal timescales over Europe. A Europe-wide analysis of the skill of the newly operational EFAS (European Flood Awareness System) seasonal streamflow forecasts (produced by forcing the Lisflood model with the ECMWF System 4 seasonal climate forecasts), benchmarked against the Ensemble Streamflow Prediction (ESP) forecasting approach (produced by forcing the Lisflood model with historical meteorological observations), is undertaken. The results suggest that, on average, the System 4 seasonal climate forecasts improve the streamflow predictability over historical meteorological observations for the first month of lead time only (in terms of hindcast accuracy, sharpness and overall performance). However, the predictability varies in space and time and is greater in winter and autumn. Parts of Europe additionally exhibit a longer predictability, up to seven months of lead time, for certain months within a season. In terms of hindcast reliability, the EFAS seasonal streamflow hindcasts are on average less skilful than the ESP for all lead times. The results also highlight the potential usefulness of the EFAS seasonal streamflow forecasts for decision-making (measured in terms of the hindcast discrimination for the lower and upper terciles of the simulated streamflow). Although the ESP is the most potentially useful forecasting approach in Europe, the EFAS seasonal streamflow forecasts appear more potentially useful than the ESP in some regions and for certain seasons, especially in winter for almost 40% of Europe. Patterns in the EFAS seasonal streamflow hindcasts skill are however not mirrored in the System 4 seasonal climate hindcasts, hinting the need for a better understanding of the link between hydrological and meteorological variables on seasonal timescales, with the aim to improve climate-model based seasonal streamflow forecasting.

## 1 Introduction

Seasonal streamflow forecasts predict the likelihood of a difference from normal conditions in the following months. Unlike forecasts at shorter timescales, which aim to predict individual events, seasonal streamflow forecasts aim at predicting long-

term (i.e. weekly to seasonal) averages. The predictability in seasonal streamflow forecasts is driven by two components of the Earth system, the initial hydrological conditions (IHC; i.e. of snowpack, soil moisture, streamflow and reservoir levels, etc.) and large-scale climate patterns, such as the El Niño-Southern Oscillation (ENSO), the North Atlantic Oscillation (NAO), the Pacific-North American (PNA) pattern and the Indian Ocean Dipole (IOD) (Yuan et al., 2015b).

The first seasonal streamflow forecasting method, based on a regression technique developed around 1910-11 in the United States, harnessed the predictability from accurate IHC of snowpack to derive streamflow for the following summer (Church, 1935). This statistical method recognised antecedent hydrological conditions and land surface memory as key drivers of streamflow generation for the following months.

Alongside the physical understanding of streamflow generation processes came technical developments, such as the creation
of the first hydrological models and the acquisition of longer observed meteorological time series, which led to the creation of the first operational model-based seasonal streamflow forecasting system. This system, called Extended Streamflow Prediction (ESP; i.e. note that ESP nowadays stands for Ensemble Streamflow Prediction, although it refers to the same forecasting method), was developed by the United States National Weather Service (NWS) in the 1970s (Twedt et al., 1977; Day, 1985). The ESP forecasts are produced by forcing a hydrological model, initialised with the current IHC, with the observed historical
meteorological time series available. The output is an ensemble streamflow forecast (where each year of historical data is a streamflow trace) for the following season(s) (Twedt et al., 1977; Day, 1985). The quality of the ESP forecasts can be high in basins where the IHC dominate the surface hydrological cycle for several months (the exact forecast quality depending on the time of year and the basin's physiographic characteristics; Wood and Lettenmaier, 2008).

In basins where the meteorological forcings drive the predictability, however, the lack of information on the future climate is
a limitation of the ESP forecasting method and might result in unskilful ESP forecasts. This drawback led to the investigation of the use of seasonal climate forecasts, in place of the historical meteorological inputs, to feed hydrological models and extend the predictability of hydrological variables on seasonal timescales (Pagano and Garen, 2006). This investigation was made possible by technical and scientific advances. Scientifically, seasonal climate forecasts were improved greatly by the understanding of ocean-atmosphere-land interactions and the identification of large-scale climate patterns as drivers of the
hydro-meteorological predictability (Goddard et al., 2001; Troccoli, 2010). This was technically implementable with the increase of computing resources, making it possible to run dynamical coupled ocean-atmosphere-land general circulation models on the global scale at high spatial and temporal resolutions (Doblas-Reyes et al., 2013). An additional technical challenge, the coarse spatial resolution of seasonal climate forecasts compared to the finer resolution of hydrological models, had to be addressed. To tackle this issue, many authors have explored different ways of downscaling climate variables for
hydrological applications (Maraun et al., 2010 and references therein).

While climate-model-based seasonal streamflow forecasting experiments are more common outside of Europe, for example for the United States (Wood et al., 2002; 2005; Mo and Lettenmaier, 2014), Australia (Bennett et al., 2016), Africa (Yuan et al., 2013), they remain limited in Europe, with a few examples in France (Céron et al., 2010; Singla et al., 2012; Crochemore et al., 2016), in Central Europe (Demirel et al., 2015; Meißner et al., 2017), in the United Kingdom (Bell et al., 2017;

Prudhomme et al., 2017) and at the global scale (Yuan et al., 2015a; Candogan Yossef et al., 2017). This is because, although the quality of seasonal climate forecasts has increased over the past decades, there remains limited skill in seasonal climate forecasts for the extra-tropics, particularly for the variables of interest for hydrology, notably precipitation and temperature (Arribas et al., 2010; Doblas-Reyes et al., 2013).

In Europe, the NAO is one of the strongest predictability sources of seasonal climate forecasts; it is associated with changes in the surface westerlies over the North Atlantic and Europe, and hence with changes in temperature and precipitation patterns over Europe (Hurrell, 1995; Hurrell and Van Loon, 1997). It was shown to affect streamflow predictability, especially during winter (Dettinger and Diaz, 2000; Bierkens and van Beek, 2009; Steirou et al, 2017), additionally to the IHC and the land surface memory. It was furthermore shown to be an indicator of flood damage and occurrence in parts of Europe (Guimarães
Nobre et al., 2017).

As the quality and usefulness of seasonal streamflow forecasts increases, their usability for decision-making has lagged behind. Translating the quality of a forecast into an added value for decision-making and incorporating new forecasting products into established decision-making chains are not easy tasks. This has been explored for many water-related applications, such as navigation (Meißner et al., 2017), reservoir management (Viel et al., 2016; Turner et al., 2017), drought-risk management
(Sheffield et al., 2013; Yuan et al., 2013; Crochemore et al., 2017), irrigation (Chiew et al., 2003; Li et al., 2017), water resources management (Schepen et al., 2016) and hydropower (Hamlet et al., 2002); but seasonal streamflow forecasts have yet to be adopted by the flood preparedness community.

The European Flood Awareness System (EFAS) is at the forefront of seasonal streamflow forecasting, with one of the first operational pan-European seasonal hydrological forecasting systems. The aim of this paper is to bridge the current gap in pan-
European climate-model-based seasonal streamflow forecasting studies. Firstly, the setup of the newly operational EFAS climate-based seasonal streamflow forecasting system is presented. A Europe-wide analysis of the skill of this forecasting system compared to the ESP forecasting approach is then presented, in order to identify whether there is any added value in using seasonal climate forecasts instead of historical meteorological observations for forecasting streamflow on seasonal timescales over Europe. Subsequently, the potential usefulness of the EFAS seasonal streamflow forecasts for decision-making
is assessed.

## 2 Data and methods

### 2.1 EFAS hydrological simulation and seasonal hindcasts

The data used in this paper include a streamflow simulation and two seasonal streamflow hindcasts (Fig. 1). Further information on these datasets is given below.

### 2.1.1 Hydrological modelling and streamflow simulation

The Lisflood model was used to produce all the simulation and hindcasts used in this paper. Lisflood is a GIS-based hydrological rainfall-runoff-routing distributed model written in the PCRaster Dynamic Modelling Language, which enables it to use spatially distributed maps (i.e. both static and dynamic) as input (De Roo et al., 2000; Van Der Knijff et al., 2010).

The Lisflood model was calibrated to produce pan-European parameter maps. The calibration was performed for 693 basins from 1994-2002 using the Standard Particle Swarm Optimisation 2011 (SPSO-2011) algorithm. The calibration was carried out for parameters controlling: snowmelt, infiltration, preferential bypass flow through the soil matrix, percolation to the lower groundwater zone, percolation to deeper groundwater zones, residence times in the soil and subsurface reservoirs, river routing and reservoir operations for a few basins. The results were validated with the Nash-Sutcliffe efficiency (NSE) for the validation

period 2003-2012. In validation [calibration], Lisflood obtained a median NSE of 0.57 [0.62]. Basins with large discrepancies between the observed and simulated flow statistics were situated mainly on the Iberian Peninsula and on the Baltic coasts (see Zajac et al., 2013 and Smith et al., 2016 for further details).

The Lisflood model is run operationally in EFAS, with the simulation domain covering Europe at a 5 x 5 km resolution. A reference simulation, called the EFAS water balance (EFAS-WB), is available on a daily time step starting from February

1990. Lisflood simulates the hydrological processes within a basin (most of which are mentioned above), starting from the previous day IHC (e.g. snow cover, storage in the upper and lower zones, soil moisture, initial streamflow, reservoir filling) and forced with the most recent observed meteorological fields (i.e. of precipitation, potential evapotranspiration and temperature; provided by the EFAS meteorological data collection centres). The observed meteorological fields are daily maps of spatially interpolated point measurements of precipitation (from more than 6000 stations) and temperature (from more than

4000 stations) at the surface level. These same data are used to produce interpolated potential evapotranspiration maps from the Penman–Monteith method (Alfieri et al., 2014). All meteorological variables are interpolated on a 5 x 5 km grid using an inverse distance weighting scheme and the temperature is first corrected using the elevation (Smith et al., 2016).

The EFAS-WB is the best estimate of the hydrological state at a given time and for a given grid point in EFAS and is thus used as initial conditions from which the seasonal hydrological forecasts are started.

### 2.1.2 Ensemble seasonal streamflow hindcasts

In this paper, two types of ensemble seasonal streamflow hindcasts are used: the Ensemble Streamflow Prediction (ESP) hindcast (hereafter referred to as ESP) and the System 4-driven seasonal streamflow hindcast [hereafter referred to as CM-SSF (climate-model-based seasonal streamflow forecast), following the notation from Yuan et al. (2015b)].

They are both initialised from the EFAS-WB, on the first day of each month, to produce a new ensemble streamflow forecast

up to a lead time of seven months (215 days), with a daily time step. Both hindcasts are generated from February 1990 for the same European domain as the EFAS-WB, at the same 5 x 5 km resolution. The unique difference between the ESP and the CM-SSF is the meteorological forcing used to drive the hydrological model, described below.

The ESP is produced by driving the Lisflood model with 20 (the number of years of data available at the time the hindcast was produced) randomly sampled years of historical meteorological observations (i.e. the same as the meteorological observations used to produce the EFAS-WB, excluding the year of meteorological observations corresponding to the year that is being forecasted). A new 20-member ESP is thus generated at the beginning of each month and for the next seven months.

The CM-SSF is produced by driving the Lisflood model with the ECMWF System 4 seasonal climate hindcast (Sys4; i.e. of precipitation, evaporation and temperature). Sys4 has a spatial horizontal resolution of about 0.7 degrees (approximately 70 km). It is re-gridded to the Lisflood spatial resolution using an inverse distance weighting scheme and the temperature is first corrected using the elevation. Sys4 is made of 15 ensemble members, extended to 51 every three months (Molteni et al., 2011). From 2011 onwards the Sys4 forecasts were run in real time and all contained 51 ensemble members. A new 15 to 51-member

CM-SSF is hence produced at the beginning of each month and for the next seven months. Operationally, the CM-SSF forecasts are currently used in EFAS to generate a seasonal streamflow outlook for Europe at the beginning of every month.

## 2.2 Hindcast evaluation strategy

For this study, monthly region specific discharge averages of the hindcasts (CM-SSF and ESP) and EFAS-WB were used. The specific discharge is the discharge per unit area of an upstream basin. For this paper, the gridded daily specific discharge was

calculated by dividing the gridded daily discharge output maps (of the hindcasts and the EFAS-WB) by the Lisflood gridded upstream area static map. Subsequently, the gridded daily specific discharge maps were used to calculate daily region averaged specific discharges (for each region in Fig. 2) by summing up the daily specific discharge values of each grid cell within a region, divided by the number of grid cells in that region. Finally, monthly specific discharge region averages were calculated for each calendar month.

The regions displayed in Fig. 2 were created by merging several basins together (basins used operationally in EFAS for the shorter timescales forecasts), while respecting hydro-climatic boundaries. They were chosen for the analysis presented in this paper for two main reasons. Firstly, they are the regions used operationally to display the EFAS seasonal streamflow outlook. Secondly, they were created in order to capture large-scale variability in the weather.

The analysis of the hindcasts was performed on monthly specific discharge (hereafter referred to as streamflow) region

averages for hindcast starting dates spanning February 1990 to November 2016 (included; approximately 27 years of data), with one to seven months of lead time. In this paper, one month of lead time refers to the first month of the forecast (e.g. the January 2017 streamflow for a forecast made on the 1[st] of January 2017). Two months of lead time is the second month of the forecast (e.g. the February 2017 streamflow for a forecast made on the 1[st] of January 2017), etc. Monthly averages were selected for the analysis presented in this paper as it is a valuable aggregation time step for decision-makers for many water-

related applications [as shown in the literature for applications such as, for example, navigation (Meißner et al., 2017), reservoir management (Viel et al., 2016; Turner et al., 2017), drought-risk management (Yuan et al., 2013), irrigation (Chiew et al., 2003; Li et al., 2017) and hydropower (Hamlet et al., 2002)].

Several verification scores were selected in order to assess the hindcasts' quality. These verification scores were chosen to cover a wide range of hindcast attributes (i.e. accuracy, sharpness, reliability, overall performance and discrimination). All of these verification scores, except for the verification score selected to look at hindcast discrimination, are the same as chosen in Crochemore et al. (2016), and are described below. The EFAS-WB streamflow simulations were used as a proxy for observation against which the seasonal streamflow hindcasts were evaluated, hence minimising the impact of model errors on the hindcasts' quality.

### 2.2.1 Hindcast accuracy

Both hindcasts (CM-SSF and ESP) were assessed in terms of their accuracy; the magnitude of the errors between the hindcast ensemble mean and the 'truth' (i.e. the EFAS-WB). For this purpose, the mean absolute error (MAE) was calculated for each region, target month (i.e. the month that is being forecast) and lead time (i.e. one to seven months). The lower the MAE, the more accurate the hindcast.

### 2.2.2 Hindcast sharpness

Both hindcasts were also assessed in terms of their sharpness; an attribute of the hindcast only, which is a measure of the spread of the ensemble members of a hindcast. In this paper, the 90% interquantile range (IQR; i.e. the difference between the 95th and the 5th percentiles of the hindcast distribution) was calculated for each region, target month and lead time. The lower the IQR, the sharper the hindcast.

### 2.2.3 Hindcast reliability

Both hindcasts were additionally assessed in terms of their reliability; the statistical consistency between the hindcast probabilities and the observed frequencies. For this purpose, the probability integral transform (PIT) diagram was calculated for each region, target month and lead time (Gneiting et al., 2007). The PIT diagram is the cumulative distribution of the PIT values as a function of the PIT values. The PIT values measure where the 'truth' (i.e. EFAS-WB) falls relative to the percentiles of the hindcast distribution. For a perfectly reliable hindcast, the 'truth' should fall uniformly in each percentile of the hindcast distribution, giving a PIT diagram that falls exactly on the 1 to 1 diagonal. A hindcast that systematically under- [over-] predicts the 'truth' will have a PIT diagram below [above] the diagonal. A hindcast that is too narrow (i.e. underdispersive; hindcast distribution smaller than the distribution of the observations) [large (i.e. overdispersive; hindcast distribution greater than the distribution of the observations)] will have a transposed S-shaped [S-shaped] PIT diagram (Laio and Tamea, 2007).

In order to compare the reliability across all regions, target months and lead times, the area between the PIT diagram and the 1 to 1 diagonal was computed for all PIT diagrams (Renard et al., 2010). The smaller this area, the more reliable the hindcast. Furthermore, to disentangle the causes for poor reliability, the spread and bias of the hindcasts were calculated for all PIT diagrams, using two measures first introduced by Keller and Hense (2011): ß-score and ß-bias, respectively. By definition, a perfectly reliable hindcast (with regards to its spread) will have a $\beta$-score of zero (to which a tolerance interval of ±0.09 was

added), whereas a hindcast that is too narrow [large] will have a negative [positive] β-score (outside of the tolerance interval). A perfectly reliable hindcast (with regards to its bias) will have a β-bias of zero (to which a tolerance interval of ±0.09 was added), whereas a hindcast that systematically under- [over-] predicts the 'truth' will have a negative [positive] β-bias (outside of the tolerance interval).

## 2.2.4 Hindcast overall performance

The hindcasts were furthermore assessed in terms of their overall performance from the continuous rank probability score (CRPS), calculated for each region, target month and lead time (Hersbach, 2000). The CRPS is a measure of the difference between the hindcast and the observed (i.e. EFAS-WB) cumulative distribution functions. The lower the CRPS, the better the overall performance of the hindcast.

In this paper, the skill of the CM-SSF is benchmarked with respect to the ESP in order to identify whether there is any added value in using Sys4 instead of historical meteorological observations for forecasting the streamflow on seasonal timescales over Europe. To this end, skill scores were calculated for the MAE, IQR, PIT diagram area and CRPS, using the following equation:

$$Skill\ score = 1 - \frac{score_{CM-SSF}}{score_{ESP}} \tag{1}$$

Skill scores were calculated for each region, target month and lead time and will be referred to as: MAESS, IQRSS, PITSS and CRPSS, respectively. Skill scores larger [smaller] than zero indicate more [less] skill in the CM-SSF compared to the ESP. A skill score of zero means that the CM-SSF is as skilful as the ESP. Note that as the ESP is not a 'naive' forecast, using it as a benchmark might lead to lower skill than benchmarking the CM-SSF against, for example, climatology.

## 2.2.5 Hindcast potential usefulness

For decision-making, the ability of a seasonal forecasting system to predict the right category of an event (e.g. above or below normal conditions) months ahead is of great importance (Gobena and Gan, 2010). In this paper, the potential usefulness of the CM-SSF and the ESP to forecast lower and higher than normal streamflow conditions within their hindcasts is assessed.

To do so, the relative operating characteristic (ROC) score, a measure of hindcast discrimination (Mason and Graham, 1999), was calculated. The thresholds selected to calculate the ROC are the lower and upper terciles of the EFAS-WB climatology for each season. They were calculated for the simulation period (February 1990 to May 2017), by grouping together EFAS-WB monthly streamflows for each month falling in a season (SON: September-October-November, DJF: December-January-February, MAM: March-April-May and JJA: June-July-August). For each season and each region a lower and upper tercile streamflow value was obtained, subsequently used as thresholds against which to calculate the probability of detection (POD) and the false alarm rate (FAR; with 0.1 probability bins) for both hindcasts, for each region, season and lead time. Finally, the area under the ROC curve, i.e. the ROC score, was calculated for both hindcasts, for each region, season and lead time. The

ROC score ranges from 0 to 1, with a perfect score of 1. A hindcast with a ROC score ≤ 0.5 is unskilful, i.e. less good than the long term average climatology which has a ROC of 0.5, and therefore not useful.

Because the ROC score was calculated from a low number of events [i.e. approximately 27 years × 3 months in each season × 1/3 (lower or upper tercile) = 27 simulated events], the hindcasts were judged skilful and useful when their ROC score ≥ 0.6 instead of 0.5. Moreover, the CM-SSF was categorised as more useful than the ESP when the CM-SSF's ROC score was at least 10% larger than the ESP's ROC score.

## 3 Results

### 3.1 Overall skill of the CM-SSF

In the first part of the results, the skill of the CM-SSF (benchmarked with respect to the ESP) is presented, in terms of the accuracy (MAESS), sharpness (IQRSS), reliability (PITSS) and overall performance (CRPSS) in the hindcast datasets. This will benchmark the added value of using Sys4 against the use of historical meteorological observations for forecasting the streamflow on seasonal timescales over Europe.

As shown by the MAESS boxplots (Fig. 3), the CM-SSF appears on average more accurate than the ESP for the first month of lead time only, for all seasons excluding spring (MAM). Beyond one month of lead time, the CM-SSF becomes on average as or less accurate than the ESP. There are however noticeable differences between the different seasons. The CM-SSF shows the largest improvements in the average accuracy compared to the ESP in winter (DJF) and for the first month of lead time. For longer lead times (i.e. two to seven months), the accuracy of the CM-SSF is on average quite similar to that of the ESP in autumn (SON) and winter, and on average lower in spring and summer (JJA). The boxplots for the CRPSS look very similar to the MAESS boxplots, the main difference being the lower average scores for two to seven months of lead time in autumn and winter (Fig. 3).

The boxplots of the IQRSS show that the CM-SSF predictions are on average as sharp as those of the ESP for the first month of lead time (slightly sharper in autumn; Fig. 3). For two to seven months of lead time, in autumn and winter, the CM-SSF predictions are on average sharper than those of the ESP, whereas in spring and summer, the CM-SSF predictions are on average slightly less sharp than the ESP predictions.

As shown by the boxplots of the PITSS (Fig. 3), the CM-SSF predictions are less reliable than the ESP prediction for all seasons and months of lead time. For the first month of lead time and all seasons, 10-20% of the ESP hindcasts and less than 5% of the CM-SSF hindcasts are reliable (Fig. 4). 40-60% of the ESP hindcasts are not reliable for the first month of lead time and all seasons due to the ensemble spread. Approximately half of these hindcasts are too large, while the other half (slightly more in autumn and winter) is too narrow. 50-80% of the ESP hindcasts furthermore under-predict the simulated streamflow for the first month of lead time and all seasons. The percentage of reliable [unreliable] ESP hindcasts increases [decreases] with lead time, as the effect of the IHC fades away. 70-90% of the CM-SSF hindcasts are too narrow for the first month of lead time and all seasons. With increasing lead time, the percentage of CM-SSF hindcasts that are too narrow [large] decreases

[increases], especially in spring. 40-50% of the CM-SSF hindcasts over-predict the simulated streamflow in spring and summer for the first month of lead time (and increasingly over-predict with longer lead times). In autumn and winter, about 70% of the CM-SSF hindcasts under-predict the simulated streamflow for the first month of lead time (and increasingly under-predict with longer lead times).

For all verification scores, the boxplots for autumn and winter are slightly smaller than for spring and summer, hinting a smaller variability in the verification scores amongst regions and target months in autumn and winter than in spring and summer. Furthermore, the presence of the boxplots above the zero line (i.e. no skill line) for all lead times suggests that the CM-SSF is more skilful than the ESP for some regions and target months, beyond the first month of lead time.

**3.2 Potential usefulness of the CM-SSF**

In the second part of the results, the potential usefulness of the CM-SSF compared to the ESP is described for decision-making. Here, potential usefulness is defined as the ability of the forecasting systems to predict lower or higher streamflows than normal, as measured with the ROC score.

Generally, either of the two forecasting systems (CM-SSF or ESP) is capable of predicting skilfully whether the streamflow
will be anomalously low or high in the coming months (Fig. 5). However, for a few seasons and regions, none of the two forecasting systems is skilful at predicting lower and/or higher streamflows than normal. This is especially noticeable in winter. For most seasons and regions, the ESP is more skilful than the CM-SSF at predicting lower and higher streamflows than normal. However, in winter for most regions and during other seasons for several regions, the CM-SSF appears more skilful than the ESP. Regions where the CM-SSF best predicts lower and higher streamflows than normal at most lead times are
summarised in Table 1 for all four seasons and the lower and upper terciles of the simulated streamflow.

**4 Discussion**

**4.1 Does seasonal climate information improve the predictability of seasonal streamflow forecasts over Europe?**

On average over Europe and across all seasons, the CM-SSF is skilful (in terms of hindcast accuracy, sharpness and overall performance, using the ESP as a benchmark), for the first month of lead time only. This means that, on average, Sys4 improves
the predictability over historical meteorological information for pan-European seasonal streamflow forecasting for the first month of lead time only. At longer lead times, historical meteorological information becomes as good as or better than Sys4 for seasonal streamflow forecasting over Europe. Crochemore et al. (2016) and Meißner et al. (2017) similarly found positive skill in the seasonal streamflow forecast (Sys4 forced hydrological model compared to an ESP) for the first month of lead time, after which the skill faded away, for basins in France and Central Europe respectively. Additionally, on average over Europe
and across all seasons, the CM-SSF is less reliable than the ESP for all lead times. This is due to a combination of too narrow and biased CM-SSF hindcasts, where the bias depends on the season that is being forecasted. As mentioned in the methods

section of this paper, the ESP is not a 'naive' benchmark, which might partially explain the limited predictability gained from Sys4.

The predictability varies per season and the CM-SSF predictions are on average sharper than and as accurate as the ESP predictions in autumn and winter beyond the first month of lead time (and increasingly sharper with longer lead times). The CM-SSF however tends to systematically under-predict the autumn and winter simulated streamflow (and increasingly under-predicts with longer lead times). In spring and summer, the CM-SSF predictions are on average less sharp and less accurate than the ESP predictions, and they tend to systematically over-predict the simulated streamflow (and increasingly over-predicts with longer lead times).

The added predictability gained from Sys4 was shown to lead to skilful CM-SSF predictions of lower and higher streamflows than normal for specific seasons and regions. The CM-SSF is more skilful at predicting anomalously low and high streamflows than the ESP in certain seasons and regions, and noticeably in winter in almost 40% of the European regions, mostly clustered in rainfall-dominated areas of Western and Central Europe. Several authors have discussed the higher winter predictability over (parts of) Europe, with examples in basins in France (Crochemore et al., 2016), Central Europe (Steirou et al., 2017), the UK (Bell et al., 2017) and the Iberian Peninsula (Lorenzo-Lacruz et al., 2011). Bierkens and van Beek (2009) additionally showed that there was a higher winter predictability in Scandinavia, the Iberian Peninsula and around the Black Sea. Our results are mostly consistent with these findings, except for Scandinavia, where the ESP is more skilful than the CM-SSF in winter. Bierkens and van Beek (2009) produced the seasonal streamflow forecast analysed in their paper by forcing a hydrological model with resampled years of historical meteorological information based on their winter NAO index. However, Sys4 has difficulties in forecasting the NAO over Europe (Kim et al., 2012), which could have led to these inconsistent results with the ones presented by Bierkens and van Beek (2009).

In spring, the CM-SSF is more skilful than the ESP at predicting lower and higher streamflows than normal beyond one month of lead time in approximately 15% of the European regions, and mostly in regions of Western Europe. This could be due to a persistence of the skill from the previous winter through the land surface memory (i.e. groundwater-driven streamflow or snowmelt-driven streamflow), as highlighted by Bierkens and van Beek (2009) for Europe, Singla et al. (2012) for parts of France, Lorenzo-Lacruz et al. (2011) for the Iberian Peninsula and Meißner et al. (2017) for the Rhine. Moreover, it could be that most of the gained predictability occurs in March, a transition month between the more predictable winter (as mentioned above) and spring, as discussed by Steirou et al. (2017). The ESP is overall more skilful than the CM-SSF at predicting the spring streamflow in snow-dominated regions (e.g. most of Fennoscandia and parts of Central and Eastern Europe). This hints the importance of the IHC (i.e. of snowpack) and the land surface memory for forecasting the spring streamflow in snow-dominated regions in Europe.

The added predictability from Sys4 for forecasting lower and higher streamflows than normal is limited in summer and autumn for most regions. The CM-SSF is more skilful at predicting anomalously low and high streamflows than the ESP in about 10-20% of the European regions during those seasons. Other studies have found similar patterns for (parts of) Europe; these include, less skill in summer than in winter overall for basins in France (Crochemore et al., 2016), less skill for the low flow

season (July to October) for basins in Central Europe (Meißner et al., 2017), negative correlations in summer and autumn seasonal streamflow forecasts in Central Europe as the influence of the winter NAO fades away (Steirou et al., 2017), and less skill overall in summer than in winter in Europe (Bierkens and van Beek, 2009). The lower CM-SSF skill for predicting lower and higher streamflows than normal in summer could additionally be due to the convective storms in summer over Europe,

which are hard to predict, and to the fact that it is the dry season in most of Europe, where rivers are groundwater fed. Therefore, in this season, the quality of the IHC controls the streamflow predictability.

While the CM-SSF is most skilful (in terms of hindcast accuracy, sharpness and overall performance, using the ESP as a benchmark) in autumn and winter and most potentially useful in winter, this does not appear to correlate with high performance in the Sys4 precipitation and temperature hindcasts [as seen on the maps of correlation for Sys4 precipitation and temperature

for all four seasons (SON, DJF, MAM and JJA) and with two months of lead time (as identified in this paper); available at https://meteoswiss.shinyapps.io/skill_metrics/]. Over Europe, the Sys4 precipitation and temperature hindcasts are the most skilful in summer and the least skilful in autumn and winter. Moreover, the regions of high CM-SSF skill for predicting lower and upper streamflows than normal do not clearly correspond to regions of high performance in the Sys4 precipitation and temperature hindcasts. These differences could be partially induced by the different benchmark used to evaluate the skill of

the CM-SSF (i.e. the ESP) compared to the one used to look at the performance of the Sys4 precipitation and temperature hindcasts (i.e. ERA Interim). However, these results clearly indicate that looking at the performance of the Sys4 precipitation and temperature hindcasts only does not give a good indication of the skill and potential usefulness of the seasonal streamflow hindcasts over Europe, and that marginal performance in seasonal climate forecasts can translate through to more predictable seasonal streamflow forecasts, and vice versa. The added predictability in the CM-SSF could be due to the combined

predictability in the precipitation and temperature hindcasts, as well as a lag in the predictability from the land surface memory. In most regions and for most seasons, at least one of the two forecasting systems (CM-SSF or ESP) is able to predict lower or higher streamflows than normal. However, in winter, the number of regions and lead times for which none of the forecasting systems are skilful increases. This could be because in winter, many regions experience weather-driven high streamflows and the performance of Sys4 is limited at this time of year (as mentioned above). In those regions, the seasonal streamflow forecasts

could be improved either by improving the IHC, through for example data assimilation, or by improving the seasonal climate forecasts.

Overall, the ESP appears very skilful at forecasting lower or higher streamflows than normal, showing the importance of IHC and the land surface memory for seasonal streamflow forecasting (Wood and Lettenmaier, 2008; Bierkens and van Beek, 2009; Yuan et al., 2015b).

**4.2 What is the potential usefulness and usability of the EFAS seasonal streamflow forecasts for flood preparedness?**

What appears like little added skill does not necessarily mean no skill for the forecast users and can in fact be a large added value for decision-making (Viel et al., 2016). The ability of a seasonal streamflow forecasting system to predict the right category of an event months ahead is valuable for many water-related applications (e.g. navigation, reservoir management,

drought-risk management, irrigation, water resources management, hydropower and flood preparedness). From the results presented in this paper, it appears that either of the two forecasting systems (CM-SSF or ESP) are capable of predicting lower or higher streamflows than normal months in advance, thanks to the predictability gained from the IHC, the land surface memory and the seasonal climate hindcast in some regions and for certain seasons.

However, as highlighted by White et al. (2017), there is currently a gap between usefulness and usability of seasonal information. What is a useful scientific finding does not automatically translate into usable information which will fit into any user's decision-making chain (Soares and Dessai, 2016). While several authors have already investigated the usability of seasonal streamflow forecasts for applications such as navigation (Meißner et al., 2017), reservoir management (Viel et al., 2016; Turner et al., 2017), drought-risk management (Sheffield et al., 2013; Yuan et al., 2013; Crochemore et al., 2017),

irrigation (Chiew et al., 2003; Li et al., 2017), water resources management (Schepen et al., 2016) and hydropower (Hamlet et al., 2002), its application to flood preparedness is still left mostly unexplored. This is partially due to the complex nature of flood generating mechanisms, still poorly studied on seasonal timescales beyond snowmelt-driven spring floods, as well as the fact that seasonal forecasts reflect the likelihood of abnormal seasonal streamflow totals, but without much skilful information on the exact timing, location and the severity of the impact of individual flood events within that season. Coughlan de Perez

et al. (2017) looked at the usefulness of seasonal rainfall forecasts for flood preparedness in Africa and highlighted the complexities behind using these forecasts as a proxy for floodiness [for discussion on floodiness see Stephens et al. (2015)]. Furthermore, decision-makers in the navigation, reservoir management, drought-risk management, irrigation, water resources management and hydropower sectors are familiar with working on long timescales (i.e. several weeks to months ahead). In contrast, the flood preparedness community is currently mostly used to working on timescales of hours to a couple of days.

The Red Cross Red Crescent Climate Centre has recently designed a new approach that harnesses the usefulness of seasonal climate information for decision-making for disaster management. This approach, called 'Ready-Set-Go!', is made of three stages. The 'Ready' stage is based on seasonal forecasts, where they are used as monitoring information to drive contingency planning (e.g. volunteer training). The 'Set' stage is triggered by sub-seasonal forecasts, used as early-warning information to alert volunteers. Finally, the 'Go!' stage is based on short-range forecasts and consists in the evacuation of people and the

distribution of aid (White et al., 2017). Using a similar approach, seasonal streamflow forecasts could complement existing forecasts at shorter timescales and provide monitoring and early-warning information for flood preparedness. Such an approach however requires the use of consistent forecasts from short to seasonal timescales. In this context, moving to seamless forecasting is becoming vital.

Soares and Dessai (2016) also identified the accessibility to the information, enhanced by collaborations and ongoing

relationships between users and producers, as a key enabler of the usability of seasonal information. International projects, such as the Horizon 2020 IMPREX (IMproving PRedictions and management of hydrological EXtremes) project (van den Hurk et al., 2016), alongside promoting scientific progress on hydrological extremes forecasting from short to seasonal timescales over Europe, gather together forecasters and decision-makers and can effectively demonstrate the added value of the integration of seasonal information in decision-making chains. The Hydrologic Ensemble Prediction EXperiment (HEPEX)

is another international initiative that brings together researchers and practitioners in the field of ensemble prediction for water-related applications. It is an ideal environment for collaboration and fosters communication and outreach on topics such as the usefulness and usability of seasonal information for decision-making.

### 4.3 Aspects for future work

In this paper, terciles of the simulated streamflow are used. However, and because the application of the EFAS seasonal streamflow forecasts is of particular relevance for flood preparedness, the evaluation of the hindcasts for lower and higher streamflow extremes (for example the 5[th] and the 95[th] percentiles respectively) would be more relevant and might give very different results. This was not done in this paper as the time period covered by the seasonal streamflow hindcasts (i.e. approximately 27 years) was not long enough for statistically reliable results for lower and higher streamflow extremes. The

limited hindcast length is a common problem in seasonal predictability studies. Increasing the hindcast length back in time could lead to more stable Sys4 hindcasts and hence to more stable and potentially skilful seasonal streamflow hindcasts (Shi et al., 2015).

Furthermore, in this paper, the hindcasts were analysed against simulated streamflow, used as a proxy for observed streamflow. This is necessary because it enables an analysis of the quality of the hindcasts over the entire computation domain, rather than

at non-evenly spaced stations over the same domain (Alfieri et al., 2014). Further work could however include carrying out a similar analysis for selected river stations in Europe, in order to account for model errors in the hindcast evaluation.

The calculation of the verification scores (excluding the ROC) was made by randomly selecting 15 ensemble members from the 51 ensemble members of the CM-SSF hindcasts, for starting dates for which the ensemble varies between 15 and 51 members (i.e. hindcasts made on the 1[st] of January, March, April, June, July, September, October and December; this is due

to the split between 15 and 51 ensemble members in the Sys4 hindcasts, as described in Sect. 2.1.2 of this paper). In order to investigate the potential impact of this evaluation strategy on the results presented in this paper, the CRPSS was calculated for 15 and 51 ensemble members of the CM-SSF hindcasts for starting dates for which 51 ensemble members are available for the full hindcast period (i.e. hindcasts made on the 1[st] of February, May, August and November). This is displayed in Fig. 6 for all hindcast starting dates, lead times (i.e. one to seven months) and regions combined. Overall, it is apparent that the impact

of this evaluation strategy on the results presented in this paper should be minimal, as all points align themselves approximately with the 1 to 1 diagonal.

The next version of the ECMWF seasonal climate forecast, SEAS5, was released in November 2017. Future work could include forcing the Lisflood model with SEAS5 and comparing the obtained seasonal streamflow hindcasts to the CM-SSF presented in this paper. This should indicate whether developments to the seasonal climate forecast translate through to better

pan-European seasonal streamflow forecasts, which is of particular interest for regions and seasons when neither the ESP nor the CM-SSF are currently skilful.

The operational EFAS medium-range streamflow forecasts are currently post-processed as a means to improve their reliability (Smith et al., 2016 and references therein). Results from this paper have shown that the CM-SSF is mostly unreliable (with

regards to the EFAS-WB) and could hence benefit from post-processing of the seasonal climate forecast. However, post-processing techniques used for the EFAS medium-range streamflow forecasts might not be suitable for the CM-SSF, as the seasonal climate forecast used for the latter should be post-processed in terms of its seasonal anomalies rather than for errors in the timing, volume and magnitude of specific events. This is currently being considered for operational implementation

within EFAS and is an active area of discussion within the EFAS user community.

For the analysis presented in this paper, the CM-SSF was benchmarked against the ESP. Several other techniques exist for seasonal streamflow forecasting, such as statistical methods using predictors ranging from climate indices to antecedent observed precipitation and crop production metrics, to mention a few (e.g. Mendoza et al., 2017; Slater et al., 2017). Further analysis could include benchmarking the CM-SSF against one or multiple statistical methods, to assess the relative benefits of

various seasonal streamflow forecasting techniques.

In this paper, the ability of both systems (CM-SSF and ESP) to forecast lower and higher streamflows than normal was explored, with several hypotheses made to link the streamflow predictability to regions' hydro-climatic processes. This includes the higher potential usefulness of the ESP in forecasting the spring streamflow in snow-dominated regions and the summer streamflow in regions where rivers are groundwater-fed. In these regions and for these seasons, the IHC and the land

surface memory drive the predictability. The CM-SSF provides an added potential usefulness in winter in the rainfall-dominated regions of Central and Western Europe, where the skill appears to persist through to spring due to the land surface memory (i.e. groundwater-driven streamflow and snowmelt-drive streamflow). While further exploration of these hypotheses is outside of the scope of this paper, future work is required to disentangle the links between the added predictability from Sys4 and the basins' hydro-climatic characteristics. For example understanding the predictability in snow-dominated basins,

arid regions and temperate groundwater-fed basins.

In this context, additional work to further disentangle and quantify the contribution of both predictability sources (seasonal climate forecasts versus IHC) to seasonal streamflow forecasting quality over Europe could be carried out by using the EPB (end point blending) method (Arnal et al., 2017).

## 5 Conclusions

In this paper, the newly operational EFAS seasonal streamflow forecasting system [producing the CM-SSF forecasts by forcing the Lisflood model with the ECMWF System 4 seasonal climate forecasts (Sys4)] was presented and benchmarked against the ESP forecasting approach (ESP forecasts produced by forcing the Lisflood model with historical meteorological observations) for the hindcast period 1990 to 2017. On average, Sys4 improves the predictability over historical meteorological information for pan-European seasonal streamflow forecasting for the first month of lead time only (in terms of hindcast

accuracy, sharpness and overall performance). However, the predictability varies per season and the CM-SSF is more skilful on average at predicting autumn and winter streamflows than spring and summer streamflows. Additionally, parts of Europe exhibit a longer predictability, up to seven months of lead time, for certain months within a season. In terms of hindcast

reliability, the CM-SSF is on average less skilful than the ESP for all lead times, due to a combination of too narrow and biased CM-SSF hindcasts, where the bias depends on the season that is being forecasted.

Subsequently, the potential usefulness of the two forecasting systems (CM-SSF and ESP) was assessed by analysing their skill in predicting lower and higher streamflows than normal. Overall, at least one of the two forecasting systems is capable of predicting those events months in advance. The ESP appears the most skilful on average, showing the importance of IHC and the land surface memory for seasonal streamflow forecasting. Nevertheless, for certain regions and seasons the CM-SSF is the most skilful at predicting anomalously low or high streamflows beyond one month of lead time, noticeably in winter for almost 40% of the European regions. This potential usefulness could be harnessed by using seasonal streamflow forecasts as complementary information to existing forecasts at shorter timescales, to provide monitoring and early-warning information for flood preparedness.

Overall, patterns in skill in the CM-SSF are however not mirrored in the Sys4 precipitation and temperature hindcasts. This hints that using seasonal climate forecast performance as a proxy for seasonal streamflow forecasting skill is not adequate and that more work is needed to understand the link between meteorological and hydrological variables on seasonal timescales over Europe.

**Acknowledgments and data availability**

L. Arnal, H. L. Cloke and J. Neumann gratefully acknowledge financial support from the Horizon 2020 IMPREX project (Grant Agreement 641811) (project IMPREX: www.imprex.eu). L. Arnal's time was additionally partly funded by a University of Reading PhD Scholarship. F. Wetterhall, C. Prudhomme and B. Krzeminski's work was supported by the EFAS computational centre in support to the Copernicus Emergency Management Service/Early Warning Systems (Flood) (contract No198702 from JRC-IES). E. Stephens is thankful for support from the Natural Environment Research Council and Department for International Development (Grant number NE/P000525/1) under the Science for Humanitarian Emergencies and Resilience (SHEAR) research programme. The data from the European Flood Awareness System are available to researchers upon request (subject to licensing conditions). Please visit www.efas.eu for more details.

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

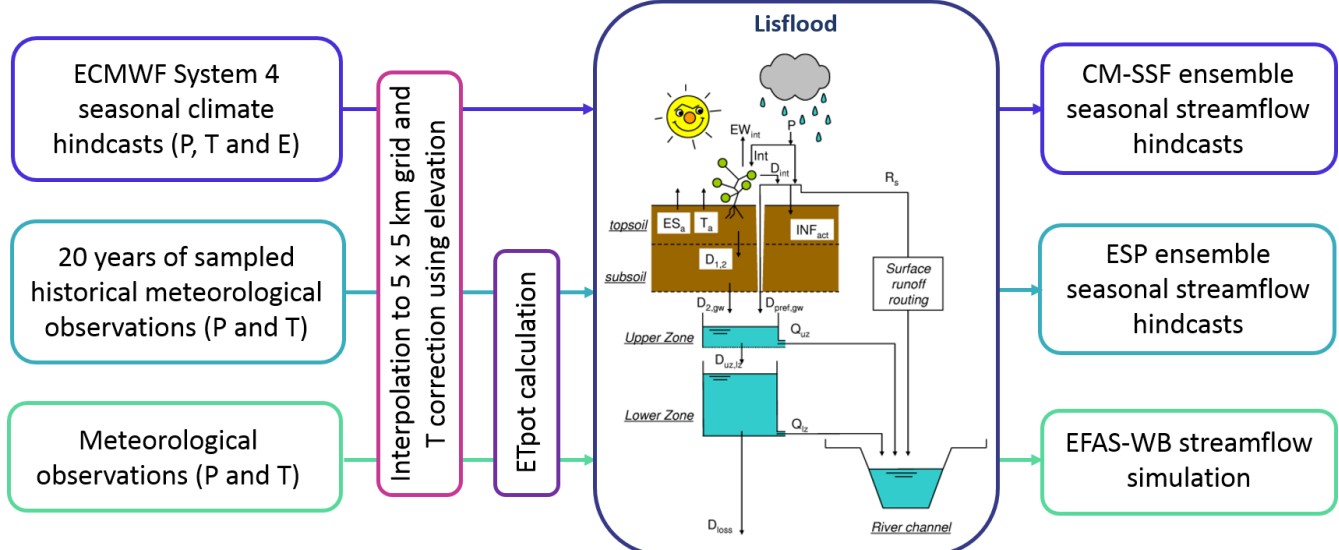

**Figure 1: Schematic of the EFAS-WB streamflow simulation and of the CM-SSF and ESP seasonal streamflow hindcasts generation. Where P: precipitation, T: temperature, E: evaporation and ETpot: potential evapotranspiration.**

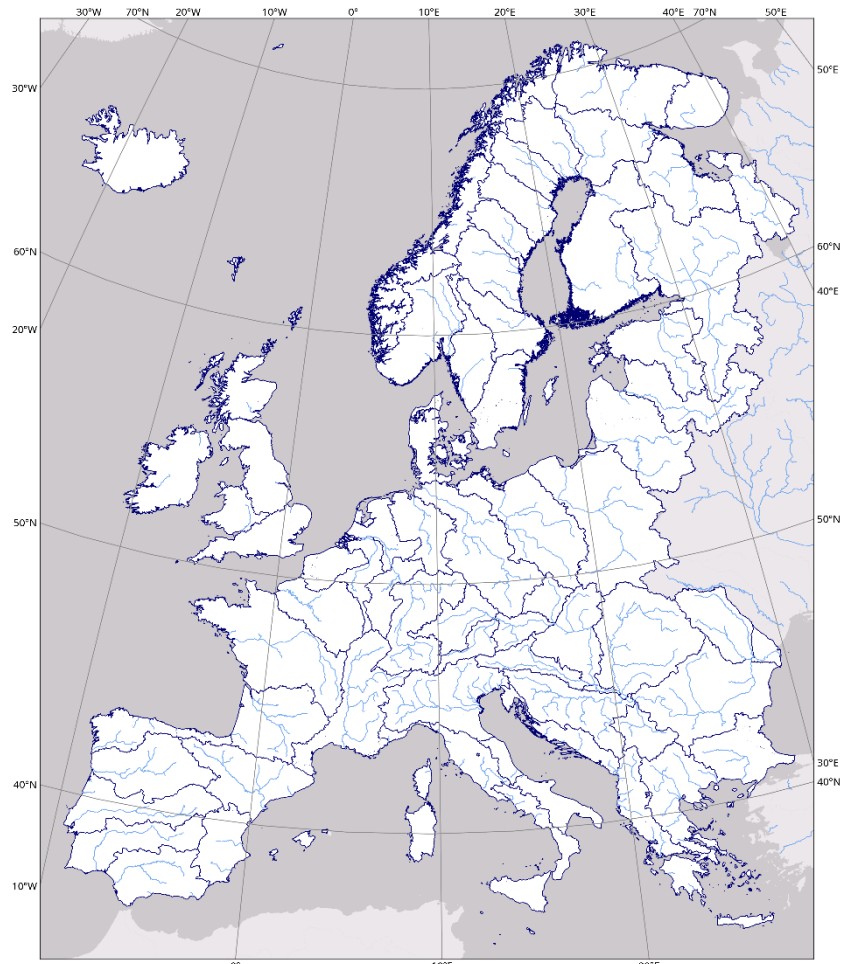

**Figure** 2**: Map of the 74 European regions (dark blue outlines) selected for the analysis of the CM-SSF and the ESP.**

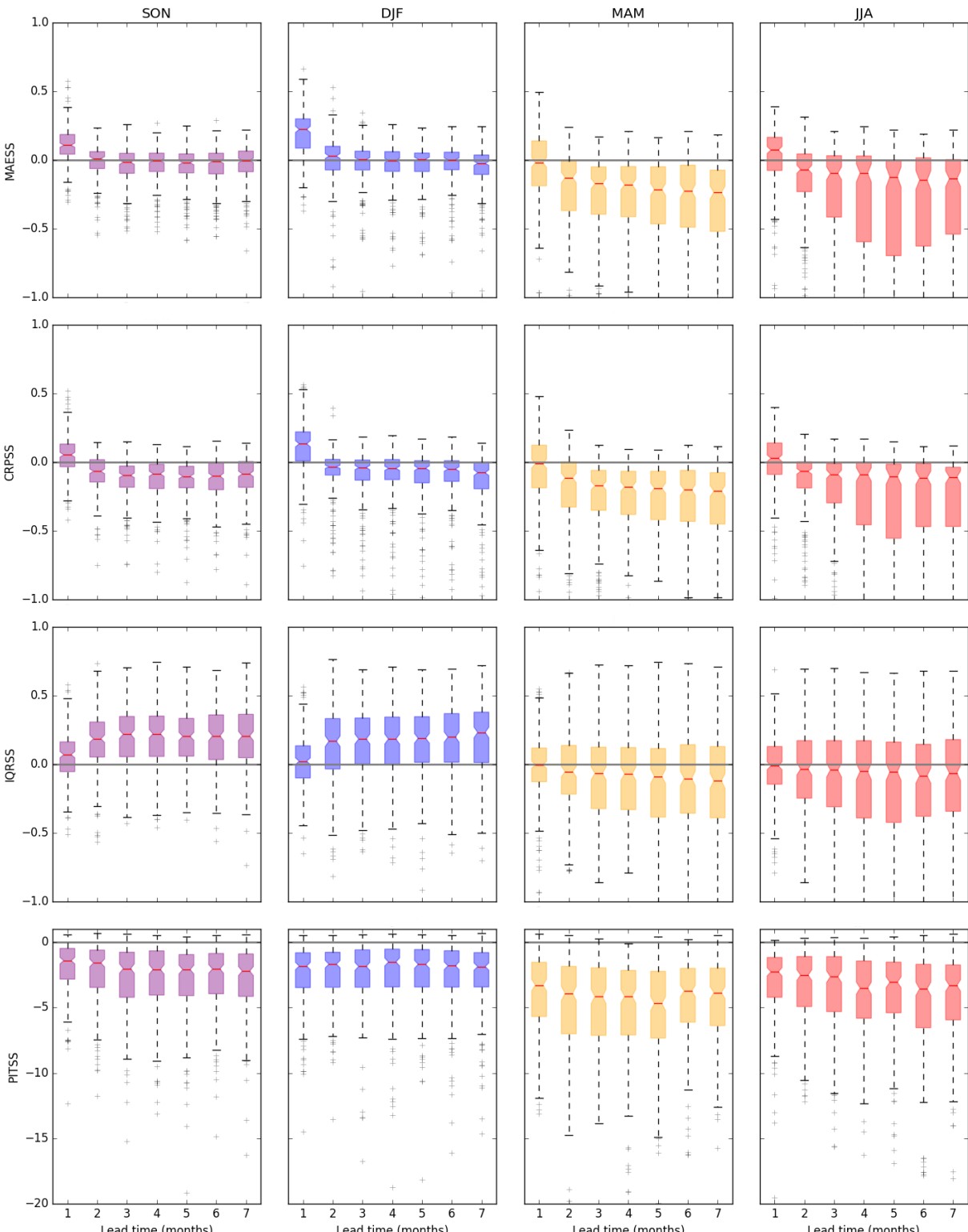

**Figure 3: Boxplots of the MAESS, CRPSS, IQRSS and PITSS (from the top to the bottom row) for all four seasons (SON, DJF, MAM and JJA from the left-most to the right-most column) as a function of lead time (i.e. one to seven months). The boxplots contain the scores for all target months falling in a given season and all 74 European regions. For all scores, values larger [smaller] than zero indicate that the CM-SSF is more [less] skilful than the ESP (benchmark). Where the skill is zero, the CM-SSF is as skilful as the ESP for the hindcast period. Note that the PITSS plots have a different y-axis scale.**

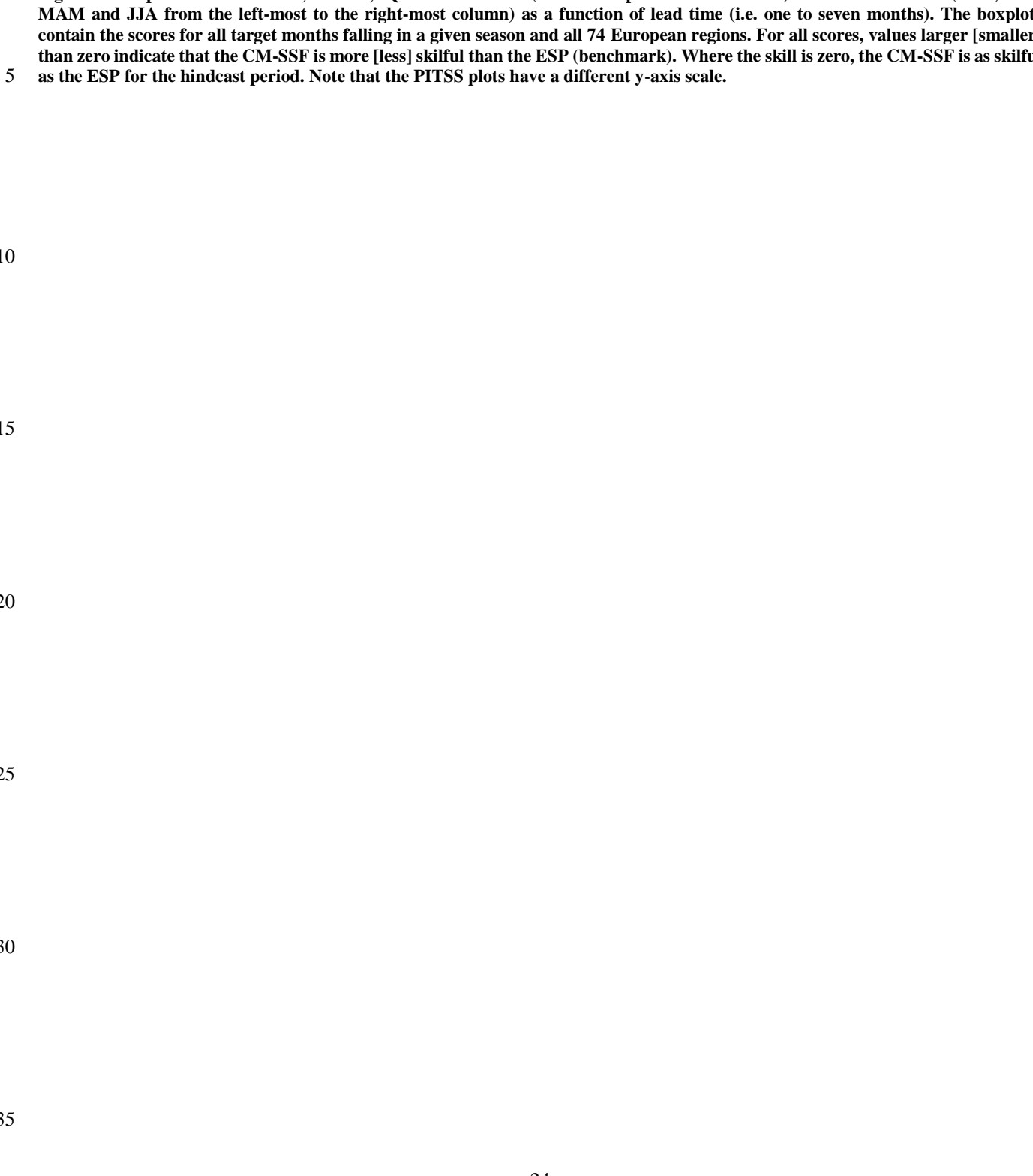

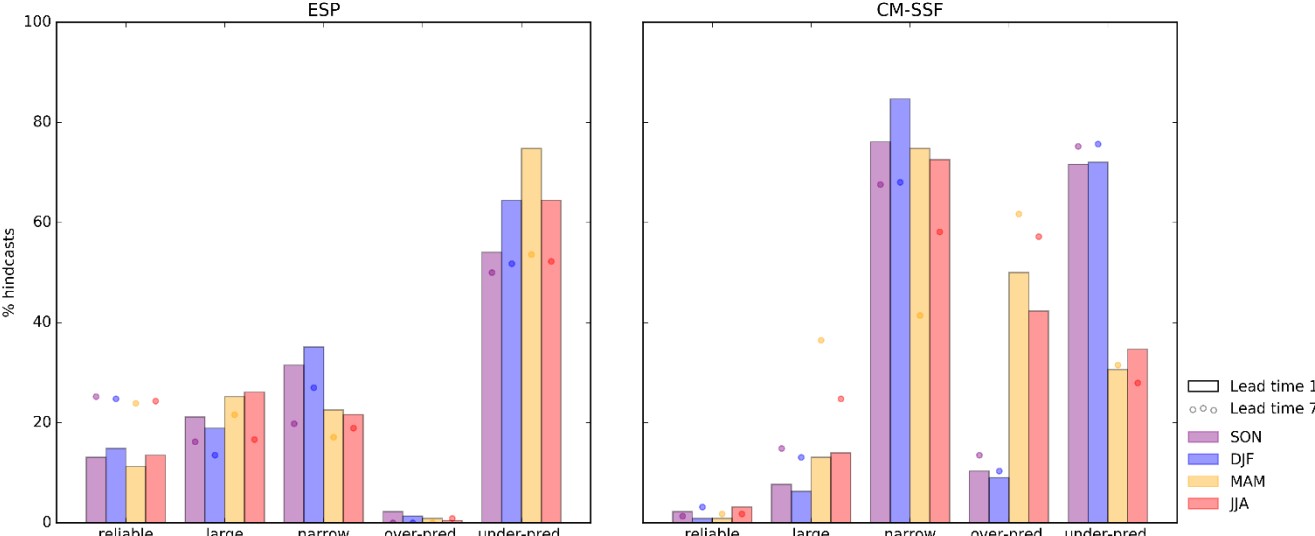

**Figure 4: Plots of the percentage of the ESP (left-hand side) and the CM-SSF (right-hand side) hindcasts falling in each reliability category (reliable - in terms of both spread and bias, too large, too narrow, over-predicting and under-predicting) for all four seasons (SON, DJF, MAM and JJA from the left-most to the right-most bars in each reliability category). The results are shown as bar charts for the first month of lead time and as circles for the seventh month of lead time. These lead times were selected for display to highlight the evolution of reliability between the first and last month of the hindcast. The percentages were calculated from hindcasts for all target months falling in a given season and all 74 European regions.**

**Table 1: Regions where the CM-SSF is more skilful than the ESP at predicting anomalously low (lower tercile; first column) or high (upper tercile; second column) streamflows for all four seasons (SON, DJF, MAM and JJA from the top to the bottom row). This is a summary of the information displayed in Fig. 5.**

|  | Lower tercile | Upper tercile |
|---|---|---|
| **SON** | - Few regions in Fennoscandia<br>- Po River basin (northern Italy)<br>- Elbe River basin (south of Denmark)<br>- Upstream of the Rhine River basin<br>- Upstream of the Danube River basin<br>- Duero River basin (Iberian Peninsula) | - Few regions in Fennoscandia<br>- Iceland<br>- Parts of the Danube River basin<br>- Segura River basin (Iberian Peninsula) |
| **DJF** | Many regions except:<br>- in most of Fennoscandia North of the Baltic Sea,<br>- parts of Central Europe. | Same as lower tercile. |
| **MAM** | - Few regions on the Iberian Peninsula<br>- Few regions in the western part of Central Europe | Same as lower tercile. |
| **JJA** | - Few regions in the United Kingdom (UK)<br>- Ireland<br>- North-western edge of the Iberian Peninsula<br>- Regions in Fennoscandia around the Baltic Sea<br>- Regions south of the North Sea | - Northern part of the UK<br>- Ireland<br>- North-western edge of the Iberian Peninsula<br>- Regions in Fennoscandia around the Baltic Sea<br>- Around the Elbe River basin<br>- Upstream of the Danube River basin<br>- Along the Adriatic Sea in Italy |

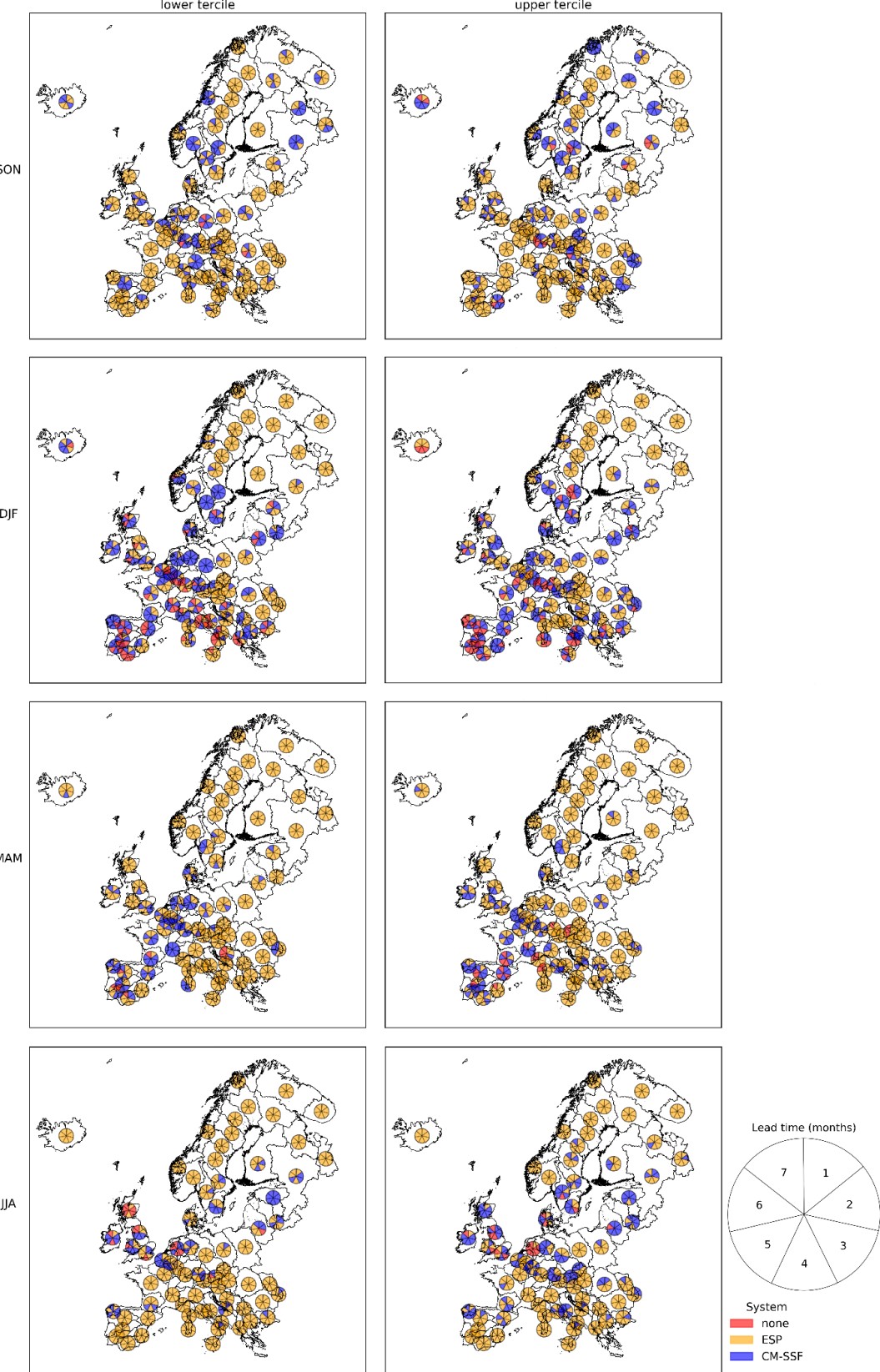

lower tercile    upper tercile

SON

DJF

MAM

JJA

Lead time (months)

7    1
6    2
5    3

System
none
ESP
CM-SSF

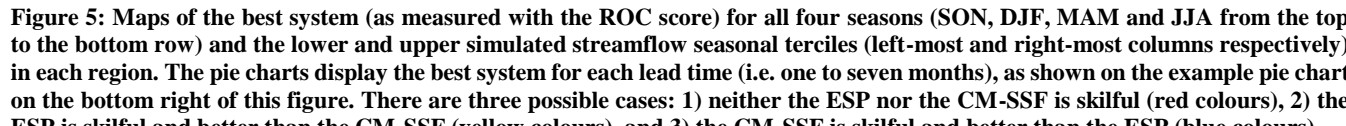

**Figure 5: Maps of the best system (as measured with the ROC score) for all four seasons (SON, DJF, MAM and JJA from the top to the bottom row) and the lower and upper simulated streamflow seasonal terciles (left-most and right-most columns respectively) in each region. The pie charts display the best system for each lead time (i.e. one to seven months), as shown on the example pie chart on the bottom right of this figure. There are three possible cases: 1) neither the ESP nor the CM-SSF is skilful (red colours), 2) the ESP is skilful and better than the CM-SSF (yellow colours), and 3) the CM-SSF is skilful and better than the ESP (blue colours).**

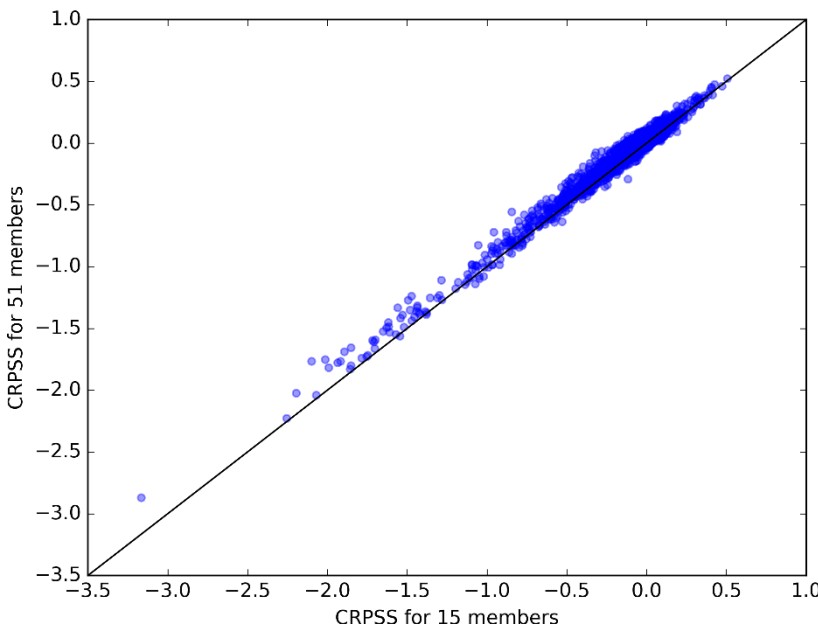

**Figure 6: CRPSS calculated for the CM-SSF against the ESP (benchmark) for hindcasts made on the 1st of February, May, August and November, all lead times (i.e. one to seven months) and all 74 European regions. The x-axis [y-axis] contains the CRPSS calculated from 15 [all 51] ensemble members of the CM-SSF.**