# Peer review of "Skilful seasonal forecasts of streamflow over Europe?"

_Hydrology and Earth System Sciences, 2017_

## Referee Comment (RC1) · Anonymous Referee #1 · 25 Oct 2017

Summary

This is a thorough and well-conceived investigation of the performance of seasonal streamflow forecasts generated by the EFAS system across Europe. The methods and verification metrics are robust and support the authors' conclusions. The manuscript is logically structured, concise and well-written, and in general I found it a very interesting read. The work sits squarely within the subject area of the special issue. I recommend that it be published after the authors consider a few minor issues for revision.

Major comments

1) This perhaps an unusual criticism, but I think the authors may have been a little too hard on their system by choosing ESP as a reference forecast. ESP is not really

a 'naive' forecast, and accordingly it is rarely used as a benchmark for performance in seasonal prediction systems. As climatology is often the default assumption by many users of forecasts, it is far more typical as a benchmark. Choosing ESP as a benchmark may have somewhat perverse results: for example, it is possible to have extremely accurate forecasts, but in cases where skill is largely due to IHCs these forecasts will not appear to be skillful (or may even be negatively skillful). This may be compounded by the use of ESP forcings that have not been cross-validated (though I may be wrong here - more information please) - i.e. it appears that an ESP hindcast from, say, 1995, could include a rainfall sequence from 1995 (a perfect forecast!) as one member of its forcing ensemble. In a small ensemble, the effect of one perfect rainfall forecast may offer some advantage to ESP forecasts compared to SyS4. I offer the following suggestions to deal with these issues:

i) When introducing the ESP reference forecasts (Section 2.1.2) please note that this is an unsually high benchmark, and why. I would also reiterate this when discussing results.

ii) If possible, cross-validate the ESP forcing ensembles (if this hasn't already been done)

iii) Note more strongly that the ROC results - which are compared to a naive benchmark - offer a more typical assessment of performance compared to CRPSS/MAESS scores calculated against ESP.

2) I would have liked more discussion of the prospects for improving reliability. Sophisticated statistical methods for calibrating ensemble climate forecasts are available to solve these issues. I would like to hear the authors' views on whether it is viable - both technically and logistically - to apply such methods within the EFAS system.

3) I thought the presentation of Figure 4 could have been improved. It's very difficult to see what the cause of the poor reliability is - bias? incorrect ensemble spread? - when all 74 basins are presented in each panel. I suggest selecting two or three

example catchments and presenting only these as case studies of possible causes of poor reliability. For comparing all 74 basins, the reliability from the PIT diagrams can be summarised with Renard et al.'s (2010) alpha index. The alpha index can be converted to a skill score, and could be added in Figure 5.

Specific (minor) comments

P1 L25-26 "Unlike forecasts at shorter timescales, they currently do not have skill to predict the exact streamflow at a specific location and time." I would argue that exact forecasts are not possible at shorter timescales either, though of course I agree that there is substantially more uncertainty at seasonal timescales.

P2 L16 "Precipitation variability was however soon identified as a major source of error in the ESP forecasts (Pagano and Garen, 2006), as this forecasting method is based on the assumption that past meteorological events are representative of future events, where each historical year has an equal likelihood of occurrence in the forecast year. As a result, the ESP forecasts are skilful as long as the weather experienced in the current year is not extraordinarily extreme compared to all the historical years of meteorological observations available (Day, 1985)." This is a little misleading, and should be recast. ESP forecasts assume that the vast majority of skill comes from IHC, and thus uses uninformative forcings. In catchments/seasons where precipitation forcings are important, this assumption does not hold, and forecasts may be inaccurate. This is distinct from out-of-sample ("extreme") events, which can occur in any system (whatever the dominant source of skill) and are (usually) difficult to predict.

P2 L29 "(Wood et al., 2002 and references therein)" This reference is fine, but quite a lot of work has been done in this area since then and it's probably worthwhile including a few more recent references.

P2 L30. There are also two papers from Greuell et al. (in review) for this special issue on a Europe-wide seasonal streamflow forecasting system.

P4 L1 "The Lisflood model was calibrated..." I'd like to hear a little more detail (perhaps a sentence or two) on the calibration method and the periods it was calibrated to. Is this calibration cross-validated?

P4 L24 "...randomly resampled..." I'm not clear on how this process is randomised. Do you simply mean 'sampled'?

P5 L24 "...hence excluding model errors from the analysis." I think this statement is too general, and could probably be removed or softened. Hydrological model errors often vary with magnitude, so different (e.g. biased) forcings can result in different hydrological error characteristics. So errors will not necessarily be 'excluded' though I understand what the authors are getting at: the main difference in forecasts and these 'observations' will be due to the forecast forcings.

P6 L5 "...The sharpness should not be looked at in isolation and 5 should be analysed together with the hindcast accuracy." I would say it's more important to check it against reliability, as sharpness can trade off reliability (e.g. a deterministic forecast is perfectly sharp, but unless it is perfect it is overconfident).

P6 L15 "...horizontal [vertical]..." this isn't really a very clear description. Forecasts that are too wide will have something like an s-shape, and forecasts that are too narrow will look something like a transposed s. The authors may like to refer to Laio and Tamea 2007, who describe these shapes in detail, for readers unfamiliar with PIT diagrams.

P14 References. A few of the papers that are listed as 'in review' are now published. Please update these.

Typos/grammar

P2 L3 "...hydrological conditions and land surface memory, as key drivers..." Delete comma

P8 L30 "...capable to predict..." should be "...capable of predicting..."

References

Greuell et al. 2017 'Seasonal streamflow forecasts for Europe' I & II, HESS special issue on Sub-seasonal to seasonal hydrological forecasting

Laio F, Tamea S. 2007. Verification tools for probabilistic forecasts of continuous hydrological variables. Hydrology and Earth System Sciences 11: 1267-1277. DOI: 10.5194/hess-11-1267-2007.

Renard B, Kavetski D, Kuczera G, Thyer M, Franks SW. 2010. Understanding predictive uncertainty in hydrologic modeling: The challenge of identifying input and structural errors. Water Resources Research 46: W05521. DOI: 10.1029/2009wr008328.

---

## Referee Comment (RC2) · Anonymous Referee #1 · 26 Oct 2017

In major comment (3), I stated "The alpha index can be converted to a skill score, and could be added in Figure 5." I meant to suggest that this could be added to Figure 3. Apologies for the error.

---

## Author Comment (AC1) · 15 Nov 2017

**Summary**

**This is a thorough and well-conceived investigation of the performance of seasonal streamflow forecasts generated by the EFAS system across Europe. The methods and verification metrics are robust and support the authors' conclusions. The manuscript is logically structured, concise and well-written, and in general I found it a very interesting read. The work sits squarely within the subject area of the special issue. I recommend that it be published after the authors consider a few minor issues for revision.**

We thank the reviewer for their very positive review and the constructive comments. Below are our responses to these comments.

**Major comments**

**1) This perhaps an unusual criticism, but I think the authors may have been a little too hard on their system by choosing ESP as a reference forecast. ESP is not really a 'naive' forecast, and accordingly it is rarely used as a benchmark for performance in seasonal prediction systems. As climatology is often the default assumption by many users of forecasts, it is far more typical as a benchmark. Choosing ESP as a benchmark may have somewhat perverse results: for example, it is possible to have extremely accurate forecasts, but in cases where skill is largely due to IHCs these forecasts will not appear to be skilful (or may even be negatively skilful). This may be compounded by the use of ESP forcings that have not been cross-validated (though I may be wrong here - more information please) - i.e. it appears that an ESP hindcast from, say, 1995, could include a rainfall sequence from 1995 (a perfect forecast!) as one member of its forcing ensemble. In a small ensemble, the effect of one perfect rainfall forecast may offer some advantage to ESP forecasts compared to SyS4. I offer the following suggestions to deal with these issues: i) When introducing the ESP reference forecasts (Section 2.1.2) please note that this is an unusually high benchmark, and why. I would also reiterate this when discussing results. ii) If possible, cross-validate the ESP forcing ensembles (if this hasn't already been done) iii) Note more strongly that the ROC results - which are compared to a naive benchmark - offer a more typical assessment of performance compared to CRPSS/MAESS scores calculated against ESP.**

Overall, we agree with the fact that the ESP is a very good benchmark and therefore a harder one to beat than, for example, climatology. One of the main reasons for choosing the ESP as a benchmark was to "identify whether there is any added value in using Sys4 instead of historical meteorological observations for forecasting the streamflow on seasonal timescales over Europe" (as mentioned on P6 L23-25).

The ESP is used as a benchmark in many seasonal forecasting papers, such as: Bazile et al. (2017), Bell et al. (2017), Candogan Yossef et al. (2017), Crochemore et al. (2016), Meißner et al. (2017) and Mendoza et al. (2017); all from this special issue.

To address the specific suggestions:

i) We will mention the superiority of the ESP to, for example, climatology, in Sections 2.2.4 and 4.1.

ii) We thank the reviewer for pointing this out. The ESP hindcast does not contain the 'perfect' year of meteorological observations as one member of its forcing ensemble. The 'perfect' year was removed to avoid increasing the ESP quality artificially (as the 'perfect' meteorological observations are not available to run the ESP in real-time). This will be clarified in Section 2.1.2 on P4 L24-25 with the following addition to the sentence "(i.e. the same as the meteorological observations used to produce

the EFAS-WB, excluding the year of meteorological observations corresponding to the year that is being forecasted)".

**2) I would have liked more discussion of the prospects for improving reliability. Sophisticated statistical methods for calibrating ensemble climate forecasts are available to solve these issues. I would like to hear the authors' views on whether it is viable - both technically and logistically - to apply such methods within the EFAS system.**

This is a very interesting point that we will add to the discussion section of this paper.

**3) I thought the presentation of Figure 4 could have been improved. It's very difficult to see what the cause of the poor reliability is - bias? incorrect ensemble spread? - when all 74 basins are presented in each panel. I suggest selecting two or three example catchments and presenting only these as case studies of possible causes of poor reliability. For comparing all 74 basins, the reliability from the PIT diagrams can be summarised with Renard et al.'s (2010) alpha index. The alpha index can be converted to a skill score, and could be added in Figure 3.**

We agree with the reviewer that it is difficult to see what the cause of the poor reliability is in Figure 4. As suggested, we will replace this figure with boxplots of the alpha index (converted to a skill score), which will be added to Figure 3. We however believe that selecting a few case studies to show the possible causes of poor reliability goes against the aim of this paper, which is to present an overall image of the performance of the EFAS seasonal streamflow hindcasts. In order to include some general information on the causes of poor reliability, we will perform a visual analysis of all the curves displayed in Figure 4 and add a summary of these causes in the paper. This could for example be summarised in a table, containing the percentage of curves in each category (i.e. narrow forecast, large forecast, under-prediction or over-prediction) for each season and lead times one and seven.

We will update the text in the methods Section 2.2 where needed.

**Specific (minor) comments**

**P1 L25-26 "Unlike forecasts at shorter timescales, they currently do not have skill to predict the exact streamflow at a specific location and time." I would argue that exact forecasts are not possible at shorter timescales either, though of course I agree that there is substantially more uncertainty at seasonal timescales.**

We thank the reviewer for this very good point. The wording is perhaps a bit misleading and we therefore propose the following alteration to this sentence: "Unlike forecasts at shorter timescales, which aim to predict individual events, seasonal streamflow forecasts aim at predicting long-term (i.e. weekly to seasonal) averages."

**P2 L16 "Precipitation variability was however soon identified as a major source of error in the ESP forecasts (Pagano and Garen, 2006), as this forecasting method is based on the assumption that past meteorological events are representative of future events, where each historical year has an equal likelihood of occurrence in the forecast year. As a result, the ESP forecasts are skilful as long as the weather experienced in the current year is not extraordinarily extreme compared to all the historical years of meteorological observations available (Day, 1985)." This is a little misleading, and should be recast. ESP forecasts assume that the vast majority of skill comes from IHC, and thus uses uninformative forcings. In catchments/seasons where precipitation forcings are important, this assumption does not hold, and forecasts may be inaccurate. This is distinct from out-of-sample ("extreme") events, which can occur in any system (whatever the dominant source of skill) and are (usually) difficult to predict.**

We thank the reviewer for this other very good point. We suggest to rephrase this sentence to: "In basins where the meteorological forcings drive the predictability, however, the lack of information on the future climate is a limitation of the ESP forecasting method and might result in unskilful ESP forecasts."

**P2 L29 "(Wood et al., 2002 and references therein)" This reference is fine, but quite a lot of work has been done in this area since then and it's probably worthwhile including a few more recent references.**

We thank the reviewer for pointing this out. We will change this reference to "(Maraun et al., 2010 and references therein)".

**P2 L30. There are also two papers from Greuell et al. (in review) for this special issue on a Europe-wide seasonal streamflow forecasting system.**

We thank the reviewer for pointing this out. We will add a reference to Greuell et al.'s paper on "Seasonal streamflow forecasts for Europe – I. Hindcast verification with pseudo- and real observations" here and in the discussion (Section 4.1), where relevant.

**P4 L1 "The Lisflood model was calibrated..." I'd like to hear a little more detail (perhaps a sentence or two) on the calibration method and the periods it was calibrated to. Is this calibration cross-validated?**

We will add the following sentences to the paper: "The calibration was performed from 1994-2002 using the Standard Particle Swarm Optimisation 2011 (SPSO-2011) algorithm. The results were validated using the Nash-Sutcliffe efficiency for the validation period 2003-2012 (see Zajac et al., 2013 and Smith et al., 2016 for more details)".

**P4 L24 "...randomly resampled..." I'm not clear on how this process is randomised. Do you simply mean 'sampled'?**

The 20 years of historical meteorological observations used for the ESP were indeed simply randomly sampled/selected from the full set of years of historical meteorological observations available (i.e. 25 in total, excluding the 'perfect' year). We will clarify this in the paper by changing "resampled" to "sampled".

**P5 L24 "...hence excluding model errors from the analysis." I think this statement is too general, and could probably be removed or softened. Hydrological model errors often vary with magnitude, so different (e.g. biased) forcings can result in different hydrological error characteristics. So errors will not necessarily be 'excluded' though I understand what the authors are getting at: the main difference in forecasts and these 'observations' will be due to the forecast forcings.**

We thank the reviewer for this comment and will change this sentence to: "The EFAS-WB streamflow simulations were used as a proxy for observation against which the seasonal streamflow hindcasts were evaluated, hence minimising the impact of model errors on the hindcasts' quality".

**P6 L5 "...The sharpness should not be looked at in isolation and should be analysed together with the hindcast accuracy." I would say it's more important to check it against reliability, as sharpness can trade off reliability (e.g. a deterministic forecast is perfectly sharp, but unless it is perfect it is overconfident).**

We will remove this sentence from the paper as we are in any case not looking at any scores in isolation in this paper.

**P6 L15 "...horizontal [vertical]..." this isn't really a very clear description. Forecasts that are too wide will have something like an s-shape, and forecasts that are too narrow will look something like a transposed s. The authors may like to refer to Laio and Tamea 2007, who describe these shapes in detail, for readers unfamiliar with PIT diagrams.**

We thank the reviewer for sharing the reference to this paper. We will remove the following sentence "A hindcast that is too narrow [wide] will have a horizontal [vertical] PIT diagram." and change it adequately using the explanations from Laio and Tamea (2007).

**P14 References. A few of the papers that are listed as 'in review' are now published. Please update these.**

The references will be updated accordingly.

**Typos/grammar** The suggestions will all be incorporated.

**References**

Bazile, R., Boucher, M.-A., Perreault, L., and Leconte, R.: Verification of ECMWF System4 for seasonal hydrological forecasting in a northern climate, Hydrol. Earth Syst. Sci. Discuss., https://doi.org/10.5194/hess-2017-387, in review, 2017.

Bell, V. A., Davies, H. N., Kay, A. L., Brookshaw, A., and Scaife, A. A.: A national-scale seasonal hydrological forecast system: development and evaluation over Britain, Hydrol. Earth Syst. Sci., 21, 4681-4691, https://doi.org/10.5194/hess-21-4681-2017, 2017.

Candogan Yossef, N., van Beek, R., Weerts, A., Winsemius, H., and Bierkens, M. F. P.: Skill of a global forecasting system in seasonal ensemble streamflow prediction, Hydrol. Earth Syst. Sci., 21, 4103-4114, https://doi.org/10.5194/hess-21-4103-2017, 2017.

Crochemore, L., Ramos, M.-H., and Pappenberger, F.: Bias correcting precipitation forecasts to improve the skill of seasonal streamflow forecasts, Hydrol. Earth Syst. Sci., 20, 3601-3618, https://doi.org/10.5194/hess-20-3601-2016, 2016.

Maraun, D., Wetterhall, F., Ireson, A. M., Chandler, R. E., Kendon, E. J., Widmann, M., Brienen, S., Rust, H. W., Sauter, T., Themessl, M., Venema, V. K. C., Chun, K. P., Goodess, C. M., Jones, R. G., Onof, C., Vrac, M., and Thiele-Eich, I.: Precipitation downscaling under climate change: Recent developments to bridge the gap between dynamical models and the end user, Reviews of Geophysics, 48, Rg3003, doi:10.1029/2009rg000314, 2010.

Meißner, D., Klein, B., and Ionita, M.: Development of a monthly to seasonal forecast framework tailored to inland waterway transport in Central Europe, Hydrol. Earth Syst. Sci. Discuss., https://doi.org/10.5194/hess-2017-293, in review, 2017.

Mendoza, P. A., Wood, A. W., Clark, E., Rothwell, E., Clark, M. P., Nijssen, B., Brekke, L. D., and Arnold, J. R.: An intercomparison of approaches for improving operational seasonal streamflow forecasts, Hydrol. Earth Syst. Sci., 21, 3915-3935, https://doi.org/10.5194/hess-21-3915-2017, 2017.

---

## Referee Comment (RC3) · Anonymous Referee #2 · 11 Dec 2017

Summary

In this manuscript, the authors assess the predictability of streamflow through the four seasons and 78 regions over Europe. Specifically they compare system 4-driven seasonal streamflow hindcast (CM-SSF) and Extended Streamflow Prediction (ESP, produced by driving Lisflood model with 20 randomly resampled years of historical meteorological observations. The main results is that the CM-SSF shows more skill when the predictions are done with 1 month of lead time for most of the regions and mainly in winter and autumn. The predictability of anomalous seasons is also assessed and the usefulness of the European Flood Awareness System (EFAS) is discussed. Results also suggest a strong dependency between initial conditions, the land surface memory and seasonal predictability for seasonal streamflow forecasting.

[Figure]

Major comments

The manuscript provides a very valuable insights on forecasting capabilities through Europe by comparing of the use of weather forecasts and historical conditions. However the contribution of the paper could be improved if the authors could establish a link between hydrological processes, climatic conditions and seasonal predictability and it is strongly recommended to include an analysis on these topics (eg. CM-SSF predictability skills in snow-dominated regions, arid regions, cold regions, etc.)

As the study region is very large, a quantitative comparison through the seasons and regions is a hard task. It is necessary to include some numbers in the result and discussion sections, perhaps there is a link between different hydro climatic regions, ungauged regions, etc. and seasonal forecast skill.

Minor comments For the different sections of the manuscript, I have some minor comments that are listed below:

Abstract The abstract needs to clarify the scientific questions and give a more quantitative assessment for the forecasting comparison (not only lead time). It is strongly suggested to include a brief methodology, materials, datasets and methods.

1. Introduction General comments

The scientific question is clear, the literature review is good but focused on European regions It would be very nice to include studies in other regions. (eg. Mendoza et al. 2014 in the Andes and Seibert et al. 2017 in southern Africa).

P1L28. Are these large-scale climatic patterns the only predictors for seasonal climatic conditions in Europe?

P2 L12. How much is high to measure forecast quality?. The response time for IHC is one of the terms that modulate the predictability of flows but also the storage capacity of watersheds in different hydrological processes (eg. Glacier, sub-surface process).

[Figure]

P3 L.15. Are there any efforts or initiatives to improve communication and outreach?

2. Data and methods

P3 L27 I strongly suggest to improve this paragraph by including information on spatial distribution, time step for modelling, quantity and quality of the data, etc.

P3 L29.Please include a citation for the Lifsflood model. The name of the hydrological model, should be included in the abstract

P4 L1, Please explain what hydrological processes were calibrated

P4 L3-14 This paragraph should be moved to section 2.1. More detail should be given to EFAS-WB (i.e., references, hydrological processes reproduced, model uncertainties, etc.) Please include forecast quality indices whenever EFAS-WB is considered the best estimate of hydrological state.

P5 L15, Did you assess of sub-monthly predictability of streamflow? Are the time scales considered enough Is that time step enough for decision makers?

P5 L 22. Please explain if the performance measures of Crochemore et al. (2016) are the ones described in the next numerals.

Sections 2.2.1 to 2.2.5 should be addressed in a Figure as a resume scheme for forecast skill.

3. Results 3.1 Overall skill of the CM-SSF In this section almost all the comparison between CM-SSF and ESP is qualitative and general. The authors could further contribute to the forecasting literature, by relating the spatial distribution of skill with physical processes and watershed type. (Eg. Do snow dominant basins shows more predictability than rainfall dominated ones?)

3.2 Potential usefulness of the CM-SSF P 8 L 31. Did the authors find specific regions where the predictability of extreme years was better/worse? If that was the case, can you please provide an explanation? P 9 L1. For most seasons (and regions?) Table 1.

This is a great contribution, downscale the regional to local scale. As commented before, it is recommended to include dominant physical processes, aridity index or other descriptors to better understand forecast skill of in different hydro climatic regimes.

4. Discussion 4.1 Does seasonal climate information improve the predictability of seasonal streamflow forecasts over Europe?

P9 – L11, In my opinion, Meiβer et al. (in review) should not be cited if there is not an accessible reference. I suggest including a DOI, or delete the reference.

P9 - L21. "The CM-SSF is more skillful in many at predicting anomalously low and high streamflows than ESP in certain season and regions". It is very difficult to qualitatively judge the skill of the predictions, but at least in terms of geographic regions, the authors could add quantitative indicators, e. g. 60% of the analyzed regions shows better performance in CM-SSF predictions rather than ESP forecast.

P10 - L8: Same comment as before.

P10 – L14-16. The analysis done in this paragraph and link to hypothesis across physical processes is desirable and should be expanded across the manuscript.

4.2 What is the potential usefulness and usability of the EFAS seasonal streamflow forecast for flood preparedness?

P11 – L23 Meumann et al. (submitted to J. Hydrometeorol.), same comment than Meiβer et al. (in review).

4.2 Aspects for future work P13 – L3-5. "The impact of this evaluation strategy in this paper should be minimal,.." But how does it impact low flows or drought predictability?

P13 -11-14. Statistical or probabilistic approaches (eg. Han and Coulibaly, 2017; Mendoza et al. 2017), should be discussed. Future work could include a different comparison, merging climate forecast with other predictors.

References

Han, S., & Coulibaly, P. (2017). Bayesian Flood Forecasting Methods: A Review. Journal of Hydrology. Mendoza, P. A., Rajagopalan, B., Clark, M. P., Cortés, G., & McPhee, J. (2014). A robust multimodel framework for ensemble seasonal hydroclimatic forecasts. Water Resources Research, 50(7), 6030-6052. Seibert, M., Merz, B., & Apel, H. (2017). Seasonal forecasting of hydrological drought in the Limpopo Basin: a comparison of statistical methods. Hydrology and Earth System Sciences, 21(3), 1611.

---

## Author Response (AR1)

**The text in bold black font are the reviewers' comments.**

The text in black font are the authors' answers to the reviewers' comments. In case of RC1, these are the same answers as were published in the interactive discussion of this paper.

The text in blue font are the specific changes made to the final manuscript as a response to the reviewers' comments.

**RC1**

**Major comments**

**1) This perhaps an unusual criticism, but I think the authors may have been a little too hard on their system by choosing ESP as a reference forecast. ESP is not really a 'naive' forecast, and accordingly it is rarely used as a benchmark for performance in seasonal prediction systems. As climatology is often the default assumption by many users of forecasts, it is far more typical as a benchmark. Choosing ESP as a benchmark may have somewhat perverse results: for example, it is possible to have extremely accurate forecasts, but in cases where skill is largely due to IHCs these forecasts will not appear to be skilful (or may even be negatively skilful). This may be compounded by the use of ESP forcings that have not been cross-validated (though I may be wrong here - more information please) - i.e. it appears that an ESP hindcast from, say, 1995, could include a rainfall sequence from 1995 (a perfect forecast!) as one member of its forcing ensemble. In a small ensemble, the effect of one perfect rainfall forecast may offer some advantage to ESP forecasts compared to SyS4. I offer the following suggestions to deal with these issues: i) When introducing the ESP reference forecasts (Section 2.1.2) please note that this is an unusually high benchmark, and why. I would also reiterate this when discussing results. ii) If possible, cross-validate the ESP forcing ensembles (if this hasn't already been done) iii) Note more strongly that the ROC results - which are compared to a naive benchmark - offer a more typical assessment of performance compared to CRPSS/MAESS scores calculated against ESP.**

We agree that the ESP is a harder benchmark to beat than, for example, climatology. However the key reason that we chose the ESP as a benchmark in this study is to "identify whether there is any added value in using Sys4 instead of historical meteorological observations for forecasting the streamflow on seasonal timescales over Europe" (as mentioned on P6 L23-25).

In addition we note that the ESP is used as a benchmark in many seasonal forecasting papers, such as: Bazile et al. (2017), Bell et al. (2017), Candogan Yossef et al. (2017), Crochemore et al. (2016), Meißner et al. (2017) and Mendoza et al. (2017); all from this special issue.

To address the specific suggestions:

i) We will mention the superiority of the ESP to, for example, climatology, in Sections 2.2.4 and 4.1.

This was added on P7 L23-24 and P10 L7-9 of the final manuscript.

ii) We thank the reviewer for pointing this out. The ESP hindcast does not contain the 'perfect' year of meteorological observations as one member of its forcing ensemble. The 'perfect' year was removed to avoid increasing the ESP quality artificially (as the 'perfect' meteorological observations are not available to run the ESP in real-time). This will be clarified in section 2.1.2 on P4 L24-25 with the following addition to the sentence "(i.e. the same as the meteorological observations used to produce

the EFAS-WB, excluding the year of meteorological observations corresponding to the year that is being forecasted)".

This was added on P5 L5-6 of the final manuscript.

**2) I would have liked more discussion of the prospects for improving reliability. Sophisticated statistical methods for calibrating ensemble climate forecasts are available to solve these issues. I would like to hear the authors' views on whether it is viable - both technically and logistically - to apply such methods within the EFAS system.**

This is a very interesting point that we will add to the discussion section of this paper.

A paragraph discussing this point has been added on P14 L11-15 of the final manuscript.

**3) I thought the presentation of Figure 4 could have been improved. It's very difficult to see what the cause of the poor reliability is - bias? incorrect ensemble spread? - when all 74 basins are presented in each panel. I suggest selecting two or three example catchments and presenting only these as case studies of possible causes of poor reliability. For comparing all 74 basins, the reliability from the PIT diagrams can be summarised with Renard et al.'s (2010) alpha index. The alpha index can be converted to a skill score, and could be added in Figure 3.**

We agree with the reviewer that it is difficult to see what the cause of the poor reliability is in Figure 4. As suggested, we will replace this figure with boxplots of the alpha index (converted to a skill score), which will be added to Figure 3. We however believe that selecting a few case studies to show the possible causes of poor reliability goes against the aim of this paper, which is to present an overall image of the performance of the EFAS seasonal streamflow hindcasts. In order to include some general information on the causes of poor reliability, we will perform a visual analysis of all the curves displayed in Figure 4 and add a summary of these causes in the paper. This could for example be summarised in a table, containing the percentage of curves in each category (i.e. narrow forecast, large forecast, under-prediction or over-prediction) for each season and lead times one and seven.

We will update the text in the methods section 2.2 where needed.

Boxplots of the alpha index (converted to a skill score) were created for all seasons and lead times and added to Figure 3. The order of the verification scores in Figure 3 was changed in order to better follow the flow of the results.

The causes for poor reliability (bias and spread) were analysed and quantified using scores introduced by Keller and Hense (2011) for all seasons, regions and lead times. These results are presented in Figure 4, as plots of the percentage of hindcasts (ESP and CM-SSF) falling within each reliability category (reliable, too large, too narrow, under- and over-predicting) for all seasons and lead time one and seven months.

The methods (sections 2.2.3 and 2.2.4), results (section 3.1) and discussion (section 4.1) sections were altered accordingly.

**Specific (minor) comments**

**P1 L25-26 "Unlike forecasts at shorter timescales, they currently do not have skill to predict the exact streamflow at a specific location and time." I would argue that exact forecasts are not possible at shorter timescales either, though of course I agree that there is substantially more uncertainty at seasonal timescales.**

We thank the reviewer for this very good point. The wording is perhaps a bit misleading and we therefore propose the following alteration to this sentence: "Unlike forecasts at shorter timescales, which aim to predict individual events, seasonal streamflow forecasts aim at predicting long-term (i.e. weekly to seasonal) averages."

This was changed on P1 L30-P2 L1 of the final manuscript.

**P2 L16 "Precipitation variability was however soon identified as a major source of error in the ESP forecasts (Pagano and Garen, 2006), as this forecasting method is based on the assumption that past meteorological events are representative of future events, where each historical year has an equal likelihood of occurrence in the forecast year. As a result, the ESP forecasts are skilful as long as the weather experienced in the current year is not extraordinarily extreme compared to all the historical years of meteorological observations available (Day, 1985)." This is a little misleading, and should be recast. ESP forecasts assume that the vast majority of skill comes from IHC, and thus uses uninformative forcings. In catchments/seasons where precipitation forcings are important, this assumption does not hold, and forecasts may be inaccurate. This is distinct from out-of-sample ("extreme") events, which can occur in any system (whatever the dominant source of skill) and are (usually) difficult to predict.**

We thank the reviewer for this other very good point. We suggest to rephrase this sentence to: "In basins where the meteorological forcings drive the predictability, however, the lack of information on the future climate is a limitation of the ESP forecasting method and might result in unskilful ESP forecasts."

This was changed on P2 L19-20 of the final manuscript.

**P2 L29 "(Wood et al., 2002 and references therein)" This reference is fine, but quite a lot of work has been done in this area since then and it's probably worthwhile including a few more recent references.**

We thank the reviewer for pointing this out. We will change this reference to "(Maraun et al., 2010 and references therein)".

This was changed on P2 L30 of the final manuscript and the reference to the paper by Maraun et al. (2010) was added to the references section of this paper.

**P2 L30 There are also two papers from Greuell et al. (in review) for this special issue on a Europe-wide seasonal streamflow forecasting system.**

We thank the reviewer for pointing this out. We will add a reference to Greuell et al.'s paper on "Seasonal streamflow forecasts for Europe – I. Hindcast verification with pseudo- and real observations" here and in the discussion (Section 4.1), where relevant.

Reviewer 2 has requested that unpublished papers 'in review' should be removed from the final manuscript. As they are still in review, the papers from Greuell et al. (in review) have therefore not been added to this final manuscript.

**P4 L1 "The Lisflood model was calibrated..." I'd like to hear a little more detail (perhaps a sentence or two) on the calibration method and the periods it was calibrated to. Is this calibration cross-validated?**

We will add the following sentences to the paper: "The calibration was performed from 1994-2002 using the Standard Particle Swarm Optimisation 2011 (SPSO-2011) algorithm. The results were

validated using the Nash-Sutcliffe efficiency for the validation period 2003-2012 (see Zajac et al., 2013 and Smith et al., 2016 for more details)".

This was added on P4 L5-12 of the final manuscript.

**P4 L24 "...randomly resampled..." I'm not clear on how this process is randomised. Do you simply mean 'sampled'?**

The 20 years of historical meteorological observations used for the ESP were indeed simply randomly sampled/selected from the full set of years of historical meteorological observations available (i.e. 25 in total, excluding the 'perfect' year). We will clarify this in the paper by changing "resampled" to "sampled".

This was changed on P5 L4 of the final manuscript.

**P5 L24 "...hence excluding model errors from the analysis." I think this statement is too general, and could probably be removed or softened. Hydrological model errors often vary with magnitude, so different (e.g. biased) forcings can result in different hydrological error characteristics. So errors will not necessarily be 'excluded' though I understand what the authors are getting at: the main difference in forecasts and these 'observations' will be due to the forecast forcings.**

We thank the reviewer for this comment and will change this sentence to: "The EFAS-WB streamflow simulations were used as a proxy for observation against which the seasonal streamflow hindcasts were evaluated, hence minimising the impact of model errors on the hindcasts' quality".

This was changed on P6 L7-9 of the final manuscript.

**P6 L5 "...The sharpness should not be looked at in isolation and should be analysed together with the hindcast accuracy." I would say it's more important to check it against reliability, as sharpness can trade off reliability (e.g. a deterministic forecast is perfectly sharp, but unless it is perfect it is overconfident).**

We will remove this sentence from the paper as we are in any case not looking at any scores in isolation in this paper.

This was removed from the final manuscript.

**P6 L15 "...horizontal [vertical]..." this isn't really a very clear description. Forecasts that are too wide will have something like an s-shape, and forecasts that are too narrow will look something like a transposed s. The authors may like to refer to Laio and Tamea 2007, who describe these shapes in detail, for readers unfamiliar with PIT diagrams.**

We thank the reviewer for sharing the reference to this paper. We will remove the following sentence "A hindcast that is too narrow [wide] will have a horizontal [vertical] PIT diagram." and change it adequately using the explanations from Laio and Tamea (2007).

This was changed on P6 L27-30 of the final manuscript and the reference to the paper by Laio and Tamea (2007) was added to the references section of this paper.

**P14 References. A few of the papers that are listed as 'in review' are now published. Please update these.**

The references will be updated accordingly.

The references were updated in the final manuscript.

**Typos/grammar.** The suggestions will all be incorporated.

These were incorporated.

**RC2**

**Major comments**

**1) The manuscript provides a very valuable insights on forecasting capabilities through Europe by comparing of the use of weather forecasts and historical conditions. However the contribution of the paper could be improved if the authors could establish a link between hydrological processes, climatic conditions and seasonal predictability and it is strongly recommended to include an analysis on these topics (eg. CM-SSF predictability skills in snow-dominated regions, arid regions, cold regions, etc.) 3.1 Overall skill of the CM-SSF. In this section almost all the comparison between CM-SSF and ESP is qualitative and general. The authors could further contribute to the forecasting literature, by relating the spatial distribution of skill with physical processes and watershed type. (Eg. Do snow dominant basins shows more predictability than rainfall dominated ones?). Table 1. This is a great contribution, downscale the regional to local scale. As commented before, it is recommended to include dominant physical processes, aridity index or other descriptors to better understand forecast skill of in different hydro climatic regimes. P10 L14-16 The analysis done in this paragraph and link to hypothesis across physical processes is desirable and should be expanded across the manuscript.**

We thank the reviewer for this constructive comment. The main aim of the paper is to give an overall overview of the skill of the EFAS seasonal streamflow hindcasts in Europe, compared to the ESP. In the discussion section of this paper (section 4.1), several hypotheses are made linking the added predictability from Sys4 for forecasting higher or lower streamflows than normal and the hydro-climatic conditions over Europe that could affect this predictability (positively or negatively). These will be clarified and extended in places.

We do agree that further results and discussion along these lines would be a great addition to this paper. In an attempt to explore this, we have split the 74 European regions used for the analysis presented in this paper in 4 climatic zones, according to the well accepted Köppen-Geiger climate classification (see map below). The latter was chosen as there is no commonly accepted hydro-climatic classification over Europe we could use. The 4 climatic zones into which the 74 regions were split are:

- Csa & Csb: warm temperate, dry and warm to hot summers
- Dfb: snow, fully humid and warm summers
- Dfc & ET: snow, fully humid and cool summers & polar, polar tundra
- Cfb: warm temperate, fully humid and warm summers

The bar chart shown below presents the percentage of European regions within each climate zone (x-axes) for which the CM-SSF is the best system at predicting anomalously low or high streamflows (lower and higher terciles in the left-hand and right-hand plots, respectively; in terms of the ROC). The results are shown for each season (SON, DJF, MAM and JJA; as indicated by the legend).

This analysis shows two main results with regards to the link between the CM-SSF predictability and the climate characteristics of the regions. More specifically, for both terciles and all seasons:

- The largest CM-SSF predictability can be found for regions in the Cfb zone, especially in winter (and summer for the upper tercile).
- The lowest CM-SSF predictability can be found for regions in the Dfb zone, especially in spring (the snowmelt season).

[Figure]

However, these results present several limitations. First, the regions used in the analysis presented in this paper are quite large. As a result, their classification in hydro-climatic zones is likely to be biased as the climate and hydrological (to a large extent) characteristics might vary greatly within a single region. Moreover, the Köppen-Geiger classification does not really capture the hydrological characteristics (i.e. infiltration, importance of groundwater reservoir, etc) of the land surface and is

more representative of the climate. Finally, the number of regions within each climate zone is highly variable and could have influenced the results shown in the bar chart.

Discussing more extensively the link between the CM-SSF predictability and the hydro-climatic characteristics of the regions from the results presented in this paper, as recommended by the editor, would require further analysis. However, as shown by the short analysis performed and described above, significant results would require a longer and more in depth analysis (i.e. by looking at the predictability for smaller river basins, by deriving basin hydro-meteorological indices and linking those with the CM-SSF predictability in those smaller basins), which we believe goes beyond the scope of this paper (i.e. to give an overall overview of the quality of the EFAS seasonal streamflow hindcasts in Europe). Therefore, we suggest to exclude any further analysis from the final manuscript and to discuss this point as an opportunity for future work in the final manuscript.

The hypotheses were clarified and extended on P10 L19 and P10 L34 – P11 L3, and the wider point was discussed on P14 L21-30 of the final manuscript.

**2) As the study region is very large, a quantitative comparison through the seasons and regions is a hard task. It is necessary to include some numbers in the result and discussion sections, perhaps there is a link between different hydro climatic regions, ungauged regions, etc. and seasonal forecast skill. P9 L21 "The CM-SSF is more skilful in many at predicting anomalously low and high streamflows than ESP in certain season and regions". It is very difficult to qualitatively judge the skill of the predictions, but at least in terms of geographic regions, the authors could add quantitative indicators, e. g. 60% of the analyzed regions shows better performance in CM-SSF predictions rather than ESP forecast. P10 L8 Same comment as before.**

We believe that this will be a great addition to the paper and will accompany the qualitative sentences in the abstract, results, discussion and conclusion sections of the paper with quantitative indicators such as the example given by the reviewer.

Following comments from the other reviewer of this paper, more quantitative results were added about the hindcasts' reliability (section 3.1). Quantitative indicators were additionally added to the abstract, discussion (P10 L18, L29 and P11 L6) and conclusion sections of the final manuscript.

**Specific (minor) comments**

**Abstract. The abstract needs to clarify the scientific questions and give a more quantitative assessment for the forecasting comparison (not only lead time). It is strongly suggested to include a brief methodology, materials, datasets and methods. The name of the hydrological model, should be included in the abstract.**

We will add research questions, mention the attributes of the hindcasts covered by the verification scores used for the analysis and the model name to the abstract.

This information was added to the abstract for the final manuscript.

**1. Introduction, general comments. The scientific question is clear, the literature review is good but focused on European regions. It would be very nice to include studies in other regions. (eg. Mendoza et al. 2014 in the Andes and Seibert et al. 2017 in southern Africa).**

In order to limit the length of the already extensive literature review and because this paper looks at the quality of the EFAS seasonal streamflow hindcasts over Europe (hence for discussion purposes), we have decided to focus the literature review on European studies. There are however references to key seasonal hydrological forecasting papers outside of Europe in the introduction of this paper.

Please see P2 L31-33 of the final manuscript for an introduction to climate-model-based seasonal streamflow forecasting experiments outside of Europe.

**P1 L28 Are these large-scale climatic patterns the only predictors for seasonal climatic conditions in Europe?**

The large-scale climate patterns mentioned here are indeed only a subset of the patterns that are currently studied and used for predicting the climate on seasonal timescales globally. We have decided to constrain our list to a few patterns that recurrently appeared in literature about seasonal hydro-meteorological forecasting.

**P2 L12 How much is high to measure forecast quality? The response time for IHC is one of the terms that modulate the predictability of flows but also the storage capacity of watersheds in different hydrological processes (eg. Glacier, sub-surface process).**

We agree with the reviewer that qualifying forecast quality of "high" is qualitative. We will improve this sentence by stating that the exact forecast quality depends on the time of year, the type and location of the basin examined.

This was added on P2 L17-18 of the final manuscript.

The storage capacity of watersheds is indeed another modulator of the flow predictability. This is however implied in the sentence "The quality of the ESP forecasts can be high in basins where the IHC dominate the surface hydrological cycle for several months" on P2 L16-17 of the final manuscript. Indeed, in basins where the storage capacity is high, the IHC are expected to have a longer impact on the output streamflow.

**P3 L15 Are there any efforts or initiatives to improve communication and outreach?**

This is a very interesting question which we have touched on slightly in the discussion section of the paper (P13 L4-9 of the final manuscript). We will however mention another international initiative, HEPEX, which has for more than a decade engaged in communication and outreach of the use of ensemble hydro-meteorological prediction for decision-making in water-related applications.

This was added on P13 L9-12 of the final manuscript.

**P3 L27 I strongly suggest to improve this paragraph by including information on spatial distribution, time step for modelling, quantity and quality of the data, etc.**

Information on spatial distribution, time step for modelling and quantity of the data is already given in sections 2.1.1 to 2.2. We will however add information on the quality of the Lisflood simulations to section 2.1.1.

This was added on P4 L9-12 of the final manuscript.

We will additionally add a sentence here clarifying that information about the data used in this paper is given below.

This was added on P3 L29-30 of the final manuscript.

**P3 L29 Please include a citation for the Lisflood model.**

The two main publications about the Lisflood model are mentioned on P4 L4 of the final manuscript and can be found in the references of this paper.

**P4 L1 Please explain what hydrological processes were calibrated.**

A list of the Lisflood hydrological processes that were calibrated will be given in section 2.1.1.

This was added on P4 L6-9 of the final manuscript.

**P4 L3-14 This paragraph should be moved to section 2.1. More detail should be given to EFAS-WB (i.e., references, hydrological processes reproduced, model uncertainties, etc.) Please include forecast quality indices whenever EFAS-WB is considered the best estimate of hydrological state.**

We believe that the current structure of section 2 is adequate and gives a nice flow to the paper. We have however added a sentence on P3 L29-30 to clarify that information about the data used in this paper is given in the following sections. More details about the hydrological processes represented within Lisflood have been given on P4 L6-9 of the final manuscript and will be referred to here (P4 L15). Information about the Lisflood simulation quality (in validation) was added on P4 L9-12.

**P5 L15 Did you assess of sub-monthly predictability of streamflow? Are the time scales considered enough? Is that time step enough for decision makers?**

The sub-monthly predictability of streamflow was not assessed for this paper. Monthly streamflow aggregations were chosen here in order to give an overview of the degradation of skill over the full seven months of lead time of the forecast. Monthly flow aggregations are valuable to decision-makers for many applications of the water sector. Indeed, in most papers cited on P3 L14-16 of the final manuscript, the authors have looked at monthly flow aggregations, with a few authors looking at three-monthly aggregations.

The use of monthly streamflow aggregations in this paper is justified on P5 L32 – P6 L3 of the final manuscript.

**P5 L22 Please explain if the performance measures of Crochemore et al. (2016) are the ones described in the next numerals.**

Here, we will clarify exactly which verification scores used in this paper were also used in Crochemore et al. (2016) and mention that these verification scores are described below.

This was changed on P6 L5-7 of the final manuscript.

**Sections 2.2.1 to 2.2.5 should be addressed in a Figure as a resume scheme for forecast skill.**

We believe that sections 2.2.1 to 2.2.5 are well explained and we do not see the added value of adding an additional figure to this paper to summarise their content.

**P8 L31 Did the authors find specific regions where the predictability of extreme years was better/worse? If that was the case, can you please provide an explanation?**

This is answered in more detail in what follows of section 3.2 and is discussed in section 4.1.

**P9 L1 For most seasons (and regions?)**

Thank you for pointing this out. It will be added here.

This was added on P9 L24 of the final manuscript.

**P9 L11 In my opinion, Meißner et al. (in review) should not be cited if there is not an accessible reference. I suggest including a DOI, or delete the reference.**

This paper is now published and the reference and corresponding in-text citations will be updated accordingly.

This was changed within the final manuscript.

**P11 L23 Neumann et al. (submitted to J. Hydrometeorol.), same comment than Meißner et al. (in review).**

This paper being currently in review, it was removed from the references.

**P13 L3-5 "The impact of this evaluation strategy in this paper should be minimal,.." But how does it impact low flows or drought predictability?**

Please note that this evaluation strategy was not used for calculating the ROC, as mentioned on P13 L27 of the final manuscript. For the latter, the full CM-SSF ensemble was used, hence not impacting low flow predictability assessed in this paper.

**P13 L11-14 Statistical or probabilistic approaches (eg. Han and Coulibaly, 2017; Mendoza et al. 2017), should be discussed. Future work could include a different comparison, merging climate forecast with other predictors.**

The comparison of the CM-SSF to statistical seasonal streamflow forecasting approaches will be mentioned in the discussion section of this paper.

A paragraph discussing this was added on P14 L16-20 of the final manuscript.

[revised manuscript text omitted]

- Po River basin (northern Italy)
- Elbe River basin (south of Denmark)
- Upstream of the Rhine River basin
- Upstream of the Danube River basin
- Duero River basin (Iberian Peninsula) | - Few regions in Fennoscandia
- Iceland
- Parts of the Danube River basin
- Segura River basin (Iberian Peninsula) |
| DJF | Many regions except:
- in most of Fennoscandia North of the Baltic Sea,
- parts of Central Europe. | Same as lower tercile. |
| MAM | - Few regions on the Iberian Peninsula
- Few regions in the western part of Central Europe | Same as lower tercile. |
| JJA | - Few regions in the United Kingdom (UK)
- Ireland
- North-western edge of the Iberian Peninsula
- Regions in Fennoscandia around the Baltic Sea
- Regions south of the North Sea | - Northern part of the UK
- Ireland
- North-western edge of the Iberian Peninsula
- Regions in Fennoscandia around the Baltic Sea
- Around the Elbe River basin
- Upstream of the Danube River basin
- Along the Adriatic Sea in Italy |

[Figure]

[Figure]

**Figure 5: Maps of the best system (as measured with the ROC score) for all four seasons (SON, DJF, MAM and JJA from the top to the bottom row) and the lower and upper simulated streamflow seasonal terciles (left-most and right-most columns respectively) in each region. The pie charts display the best system for each lead time (i.e. one to seven months), as shown on the example pie chart on the bottom right of this figure. There are three possible cases: 1) neither the ESP nor the CM-SSF is skilful (red colours), 2) the ESP is skilful and better than the CM-SSF (yellow colours), and 3) the CM-SSF is skilful and better than the ESP (blue colours).**

[Figure]

**Figure 6: CRPSS calculated for the CM-SSF against the ESP (benchmark) for hindcasts made on the 1st of February, May, August and November, all lead times (i.e. one to seven months) and all 74 European regions. The x-axis [y-axis] contains the CRPSS calculated from 15 [all 51] ensemble members of the CM-SSF.**

---

## Author Response (AR2)

P2 L18: Basin location is not a driver to affect the forecast quality. It is rather the 'physiographic characteristics of the basin', which could occasionally have a spatial structure. Please remove 'location'.

'the type and location of the basin' was changed to 'the basin's physiographic characteristics' on P2 L18 of the final manuscript.

P4 L10: Please mention the number of stations used in the calibration/evaluation. In addition I believe that it is misleading to imply that NSE > 0 has an explanatory power (NSE > 0 means that the model is predicting better than the mean discharge). I would suggest you mention the mean and median NSE values for the calibration/evaluation stations instead.

The number of stations used in the calibration was added on P4 L5 of the final manuscript. The median NSE obtained for the calibration and validation were mentioned instead on P4 L10.

Figure 1 as well as its caption were additionally slightly modified in order to be more accurate.

**Skilful seasonal forecasts of streamflow over Europe?**

Louise Arnal1,2, Hannah L. Cloke1,3,4,5, Elisabeth Stephens1, Fredrik Wetterhall2, Christel Prudhomme2,6,7, Jessica Neumann1, Blazej Krzeminski2 and Florian Pappenberger2

[revised manuscript text omitted]